# The State of the Art and Challenges of In Vitro Methods for Human Hazard Assessment of Nanomaterials in the Context of Safe-by-Design

**DOI:** 10.3390/nano13030472

**Published:** 2023-01-24

**Authors:** Nienke Ruijter, Lya G. Soeteman-Hernández, Marie Carrière, Matthew Boyles, Polly McLean, Julia Catalán, Alberto Katsumiti, Joan Cabellos, Camilla Delpivo, Araceli Sánchez Jiménez, Ana Candalija, Isabel Rodríguez-Llopis, Socorro Vázquez-Campos, Flemming R. Cassee, Hedwig Braakhuis

**Affiliations:** 1National Institute for Public Health & the Environment (RIVM), 3721 MA Bilthoven, The Netherlands; 2Univ. Grenoble-Alpes, CEA, CNRS, SyMMES-CIBEST, 17 rue des Martyrs, 38000 Grenoble, France; 3Institute of Occupational Medicine (IOM), Edinburgh EH14 4AP, UK; 4Finnish Institute of Occupational Health, 00250 Helsinki, Finland; 5Department of Anatomy, Embryology and Genetics, University of Zaragoza, 50013 Zaragoza, Spain; 6GAIKER Technology Centre, Basque Research and Technology Alliance (BRTA), 48170 Zamudio, Spain; 7LEITAT Technological Center, 08225 Barcelona, Spain; 8Instituto Nacional de Seguridad y Salud en el Trabajo (INSST), 48903 Barakaldo, Spain; 9Institute for Risk Assessment Sciences (IRAS), Utrecht University, 3584 CS Utrecht, The Netherlands

**Keywords:** nanomaterials, safe-by-design, hazard testing, in vitro methods, SAbyNA, advanced materials

## Abstract

The Safe-by-Design (SbD) concept aims to facilitate the development of safer materials/products, safer production, and safer use and end-of-life by performing timely SbD interventions to reduce hazard, exposure, or both. Early hazard screening is a crucial first step in this process. In this review, for the first time, commonly used in vitro assays are evaluated for their suitability for SbD hazard testing of nanomaterials (NMs). The goal of SbD hazard testing is identifying hazard warnings in the early stages of innovation. For this purpose, assays should be simple, cost-effective, predictive, robust, and compatible. For several toxicological endpoints, there are indications that commonly used in vitro assays are able to predict hazard warnings. In addition to the evaluation of assays, this review provides insights into the effects of the choice of cell type, exposure and dispersion protocol, and the (in)accurate determination of dose delivered to cells on predictivity. Furthermore, compatibility of assays with challenging advanced materials and NMs released from nano-enabled products (NEPs) during the lifecycle is assessed, as these aspects are crucial for SbD hazard testing. To conclude, hazard screening of NMs is complex and joint efforts between innovators, scientists, and regulators are needed to further improve SbD hazard testing.

## 1. Introduction

The rapid expansion of the field of nanotechnology and its ever-growing number of applications has created a challenge for toxicologists and risk assessors. The continuous uncertainties surrounding nanomaterial (NM) safety, as well as the pace at which new NMs are developed, call for a more prevention-oriented strategy. The Safe-by-Design (SbD) concept is increasingly applied within the field of nanotechnology, as can be seen by the high number of EU funded nano-projects addressing SbD over the past years [1], and by its adoption in the EU Chemical Strategy for Sustainability as a strategy to meet the EU Green Deal ambitions [2,3].

SbD aims to reduce the human and environmental risk of a substance throughout its entire life cycle by minimizing or eliminating the hazard and/or by reducing exposure [4]. The concept of SbD consists of three pillars: safer materials and products, safer production, and safer use and end-of-life. For NMs, these were first described in the NanoReg2 project [5], and later in an internationally accepted working description of the OECD Safe Innovation Approach Report [6]. In practice, SbD is a two-step process: the first step is an early hazard and/or risk screening during the design phase of the innovation process of a new substance, NM, or product [7,8]. The second step is to take actions (SbD interventions) to reduce or minimize hazard, exposure, or both. 

For NMs and nano-enabled products (NEPs), SbD interventions can be achieved in different ways. One option is to modify a NM in order to improve its safety profile. For example, Xia et al. (2011) showed that doping ZnO nanoparticles with iron reduces the shedding of harmful ions and reduces the toxicity of the particles upon pulmonary exposure [9]. Another example of a SbD intervention is applying a surface treatment to minimise NM biological reactivity, as has been successfully achieved for nano-SiO_2_ by adding silanol groups to the silica surface [10]. Reducing exposure is also a fundamental part of SbD and can be achieved by implementing procedural changes such as working in closed systems or using wet synthesis methods [5]. Reduced release and therefore minimized exposure can also be achieved by altering the design of the NEP, for example by improving the immobilization between the NM and the matrix, as was conducted for silver NMs onto cotton fabrics [11].

The above-mentioned examples can only be achieved after first assessing hazard and risk early in the innovation process, and then using this knowledge to integrate safety into the design of the NM, NEP, or production process. For many NMs, and especially for novel ones, hazards are largely unknown [12], and cannot be predicted only based on physicochemical (PC) characterisation. Therefore, carrying out suitable hazard testing at the early stages of product development is of utmost importance for SbD applicability. SbD hazard testing aims to identify hazard warnings in the early stages of the innovation process using simple in vitro methods. Once a product is designed and produced, the manufacturer should comply to the regulations and perform hazard and risk assessment accordingly. 

Many strategies and frameworks for hazard assessment of NMs in the context of SbD have been proposed in recent years [13,14,15,16,17,18], some proposing specific in vitro assays, and some based only on a selection of toxicological endpoints to consider. However, no comprehensive investigation of the suitability of currently available in vitro assays for such strategies has been conducted thus far.

From previous studies on the mechanisms of action of NMs, it is known that transformation (e.g., dissolution), reactivity, inflammation, cytotoxicity, and genotoxicity are among the most important parameters and endpoints to evaluate when assessing the hazard of a NM, and therefore these are suggested to be measured in many available strategies and frameworks [13,19,20,21]. Selecting in vitro tests suitable for SbD hazard testing is not trivial. Only a few OECD test guidelines and ISO standards are available specifically for testing NMs. Due to the interfering behaviour of NMs with their surrounding environment and with the assay readout, routinely used toxicity assays (i.e., those used to test soluble chemicals) may prove unsuitable or may require optimizations and inclusion of extra controls [22]. In contrast with hazard assessment for soluble chemicals, NM testing requires additional steps, such as dispersion protocols and determining the dose delivered to the cells in submerged cell culture experiments [23]. Specifically for the purpose of SbD hazard testing, since it is performed early in the development of a NM/NEP, assays will not only have to be compatible with the NM to be tested, but should preferably also be fast, cost-effective, and able to correctly indicate hazard warnings.

This work provides a practical and critical evaluation of the suitability of most frequently used in vitro toxicity assays and the challenges for their use in NM SbD hazard testing. For this purpose, criteria for the suitability of methods for application in a SbD hazard testing strategy are established, leading to an evaluation of the methods currently in use for the parameters and endpoints identified as important for the mechanisms of action of NMs. This work is conducted under the umbrella of the Horizon2020 project SAbyNA which aims to develop a user-friendly platform for industry with optimal workflows to support the development of SbD NMs and NEPs. For this purpose, existing resources, such as in vitro assays are identified, distilled, and streamlined. This state of the art and evaluation of in vitro assays for SbD applicability can be used as an outlook for innovators, regulators, industry, and scientists of how early hazard testing of NMs and NEPs can be put into practice to eventually contribute to the design of SbD NEPs.

## 2. Criteria

A set of performance criteria is proposed to evaluate the suitability of in vitro methods for SbD hazard testing. The criteria were adapted from the widely used Good In Vitro Method Practices (GIVIMP) [24] and tailored to suit SbD hazard testing for NMs specifically. Figure 1 shows several key considerations for assay selection for SbD hazard testing.

**Predictive:** The first criterium is that an in vitro assay should be sufficiently predictive of the in vivo situation. This comparison is preferably made with human data, or alternatively using animal data. SbD hazard testing is carried out in an early stage of product development and is considered a first screening. The aim of SbD hazard testing is to detect **early hazard warnings** and not to derive a point of departure for risk assessment. Therefore, assays are sufficiently predictive for SbD hazard testing when they are able to indicate hazard warnings. Assays that are able to **accurately rank** NMs/NEPs based on their hazard potency are of extra value for SbD hazard testing, as this will allow comparison of candidate NMs and comparison with benchmark NMs.

Predictivity can be assessed by looking into the prediction accuracy of the assay. An assay’s **accuracy** to predict in vivo effects is a combination of its **sensitivity** and **specificity**. Sensitivity is the ability of the assay to detect true positives and specificity is the ability of the assay to detect true negatives.

**Simple and cost effective:** Simplicity and cost effectiveness are key for SbD hazard testing since these assays are to be performed in an early stage of NM/NEP product development. Ideally, an assay should be easy to perform, time-efficient and cost-effective.

**Robust:** An assay should give consistent and repeatable results between experimental repetitions and between different labs.

**Compatible:** An assay should preferably be compatible with a wide range of NMs, or at least its compatibility domain should be identified. Assays with optimized protocols specifically for NMs are preferred.

**Readiness:** Methods that are considered ‘ready to use’ and already standardized or (pre-)validated for NMs are prioritized.

## 3. Challenges of Testing NMs In Vitro for SbD Applicability

NMs are particulate matter, making NM in vitro testing by default more challenging than testing soluble chemicals. Several additional aspects need considering when testing NMs in vitro, including determining the behaviour of the NM in exposure medium, selecting a dispersion protocol to create stable suspensions which preferably mimic human exposure as much as possible, and assessing the potential interference of the NM with the assay components or optical readouts. Furthermore, elaborate characterization of the NM is required [25], but this will not be discussed further in this review. The fact that SbD hazard testing needs to be as simple as possible creates an important predicament that needs addressing. An overview of key aspects that should be taken into account when performing SbD hazard testing of NMs is shown in Figure 2. The most important aspects are discussed below.

### 3.1. Choice of Dispersion Protocol

Classically, in vitro toxicity evaluation is performed in cultured cells maintained in submerged conditions. To ensure reproducible and controlled exposure from one replicate to another, stable suspensions of well-dispersed NMs are prepared, sometimes requiring energy input to disrupt particle agglomerates. For SbD hazard testing, a prerequisite for the suitability of a dispersion procedure is that the SbD properties (e.g., coatings or surface treatments) of the dispersed NM/NEP are preserved, and that the resulting dispersed NM/NEP is relevant for human exposure in terms of size and other physicochemical properties.

The most commonly used dispersion procedure for NMs is via sonication, using either an ultrasonic bath, a probe, or a cup-type sonicator [26,27]. For SbD hazard testing, sonication would not be the preferred option in some specific cases. For instance, sonication can break multi-walled carbon nanotubes (MWCNT), causing a reduction of their length [27], and therefore leading to different toxicity profiles than the MWCNT that humans would be exposed to. Sonication has been used to produce MWCNTs with different lengths from a same initial batch of MWCNTs [28,29], and Hadrup et al. (2021) concluded that the length of the MWCNT is a major determinant of its toxicity [29]. Therefore, when assessing hazard properties of CNTs in vitro, sonication should be limited as much as possible, and in case sonication is used, NM physicochemical properties should be verified to ensure they still maintain exposure-relevant characteristics.

Another example where sonication would have to be carefully considered is when testing specific synthetic amorphous silicas, which are in some cases intentionally produced as agglomerates. Sonication can disrupt most of the agglomerates, reducing the overall hydrodynamic diameter, which constitutes a substantial modification of the initial material [30]. In an inhalation exposure scenario, a person would be exposed to these agglomerates, and therefore sonication would not be the preferred option. However, there are indications that these agglomerates disintegrate in the intestine [31]. In the case of ingestion, the gut epithelium is exposed to nano-sized silica, and sonication would result in an exposure-relevant material.

Dissolution is a major determinant of the toxicity of some NMs (e.g., release of toxic metal ions such as silver or copper ions) and decreasing the NM dissolution potential can be considered a SbD intervention. Sonication has been shown to enhance the dissolution of some metallic NMs, such as Cu, Mn, and Co [32], and the dissolution can be further increased when proteins are present in the solution during sonication [33]. Thus, dispersion procedures that involve sonication of NMs, especially in a medium that contains proteins, such as the procedure optimized within the Nanogenotox project [34], should be carefully considered in view of exposure scenarios when testing NMs that can potentially dissolve.

Some NMs are designed as core-shell structures (e.g., quantum dots (QDs)) where the shell reduces dissolution and leaching of potentially toxic elements from the core. The design of more robust shells, used as a SbD intervention, reduces the QDs dissolution rate and thereby their toxicity [35,36,37]. However, the core-shell boundary is a region of fragility and sonication could promote shell fragmentation and the release of core contents. Thus, sonication could result in a reduced effect of the SbD intervention, potentially resulting in an overestimation of toxicity in SbD hazard testing. Therefore, for core-shell NMs, sonication should not be recommended, unless humans are exposed to fragmented QDs.

Coating NMs with surface ligands or grafting them on an inert matrix such as cellulose has also been tested as a SbD intervention to produce safe(r) photocatalytic paints containing TiO_2_ NMs. Coating TiO_2_ NMs with polyethylene glycol (PEG), poly(acrylic acid) (PAA), or 3,4-dihydroxy-L-phenylalanine (DOPA) increased the stability of the doped paint and its resistance to weathering and abrasion, while their grafting on cellulose fibres enhanced their photocatalytic properties, thereby allowing for the reduction of the amount of NMs necessary to reach efficient photocatalysis [38]. Again, sonicating these surface-coated TiO_2_ NMs or TiO_2_-containing composites might lead to the reduction of the effect of SbD interventions. In addition to that, extensive sonication has been shown to alter the zeta potential of TiO_2_ and CeO_2_ NMs [39,40] and to cause re-agglomeration of Cu or Mn NMs [33,41]. It should in each case be investigated what the exposure-relevant form of the NM or NEP is.

Moreover, samples could be contaminated by the release of Al and Ti from the sonication probe upon extensive sonication, potentially leading to toxicity [30,42]. Finally, extensive sonication of NMs in a growth medium containing proteins or in water with bovine serum albumin (BSA) (as in the procedure optimized within the Nanogenotox project [34]) could promote the degradation of proteins, leading to the formation of large aggregates of degraded proteins [43].

To conclude, for SbD hazard testing sonication should be carefully considered, and exposure relevancy should always be kept in mind. If exposure-relevant and stable dispersions in the exposure medium are obtained using simple methods such as vortexing, dispersion via sonication might not be needed. In the case of NMs that quickly agglomerate and form large clumps, more controlled sonication methods might be appropriate. For example, a protocol using minimum material-specific energy to reach a stable dispersion as described by DeLoid et al. (2017) could be used [23]. NMs should subsequently be characterized to ensure that no PC changes were produced that deviate from the exposure-relevant material. The PC properties of the NM tested should reflect the exposure conditions, whether it be the pristine NM with SbD interventions or the agglomerated NM. However, it has to be noted that unstable suspensions could lead to difficulties with reproducibility and/or interferences with the assay readout.

Finally, it might be recommended to also perform in vitro assays after extensive sonication, as this might be required for regulatory risk assessment. By doing this, the first steps towards compiling a dossier for regulatory compliance are made, and this might already indicate if any issues can be foreseen for market entry. Additionally, extensive sonication may provide a worst-case scenario in in vitro assays, which could fit in a precautionary approach. OECD guidance on sample preparation [44] is currently being revised to include considerations for the choice of a specific dispersion protocol rather than applying extensive sonication by default.

For SbD hazard testing sonication should be carefully considered, and exposure relevancy should always be kept in mind. If exposure-relevant and stable dispersions in the exposure medium are obtained using simple methods such as vortexing, dispersion via sonication might not be needed. In the case of NMs that quickly agglomerate and form large clumps, more controlled sonication methods might be appropriate. NMs should subsequently be characterized to ensure that no PC changes were produced that deviate from the exposure-relevant material.

### 3.2. Influence of Medium Components

Supplementation of cell-culture medium with serum (i.e., foetal calf serum (FCS) and foetal bovine serum (FBS)) is common practice in cell culture procedures as it is required for cell growth and maintenance. When exposing the NMs in a test medium, constituents of the medium including proteins, amino acids, lipids, and sugars adsorb on the surface of NMs, leading to the formation of the so-called biomolecular corona [45]. This corona is highly dynamic and may change upon changing the composition of the test medium [46,47]. This dense layer of biological molecules can modify NM toxicity in several ways. Firstly, it could do so by masking the surface reactive sites of the NM [48]. Secondly, serum may stabilize the NM dispersion, leading to a lower dose delivered to the cells in in vitro assays, as has been shown for TiO_2_ NMs for example [49]. Lastly, a biomolecular corona may reduce NM surface energy, and thereby its cellular uptake via adhesion-induced endocytosis, as has been shown for SiO_2_ NMs [50,51].

These effects are clear when comparing the toxicity of NMs tested with and without serum. For instance, the cytotoxic potency of polystyrene NMs was found to decrease 2-fold when the exposure medium contained serum [52]. Similarly, the cytotoxicity of SiO_2_ NMs decreased up to 92%, and pro-inflammatory response decreased up to 87% when cells were exposed in medium with serum [53]. In addition, the species of origin of the serum could lead to different responses [54]. Addition of bovine serum albumin (BSA), a protein often used to help stabilize dispersions, has also been reported to reduce cytotoxicity [55,56].

In short, the addition of serum and BSA to the exposure medium may lead to lower toxicity in in vitro assays. Which approach is most suitable for SbD hazard testing should be explored further. Since SbD hazard testing is mostly focused on detecting hazard warnings, it could be argued that testing without serum is more appropriate, as it ensures a higher sensitivity. Additionally, when testing serum-free, a worst-case scenario could be mimicked without the protective effect of serum on NM-cell interaction. On the other hand, testing with serum is the more realistic approach as humans are rarely exposed to NMs without a biomolecular corona. Eventually, the route of (potential) human exposure should be taken into account when selecting an exposure protocol, as systemically injected NMs will immediately be covered by serum proteins, whereas inhaled NMs will come in contact with epithelial lining fluids, which contains a different set of biomolecules. For SbD hazard testing, exposure relevancy is important, and a biocorona could be applied which corresponds to the route of exposure, such as lung-lining fluid for pulmonary exposure. In the context of exposure relevance, and in the context of the 3Rs (replacement, reduction, and refining), the use of human serum or serum-free alternatives may be favoured over FBS.

### 3.3. Determining Dose Delivered to Cells

A particular challenge when testing NMs in vitro in adherent, submerged cell cultures is determining the delivered dose, i.e., the amount of material that reaches the cells. Settling of NMs depends on their density, size, and the properties of the cell-culture medium, as well as on their agglomeration state [57]. The latter is again influenced by the dispersion method used [58].

Determining the delivered dose in an in vitro experiment is an absolute requirement, even when performing a simple hazard screening for the purpose of SbD, and when performing high throughput screening (HTS) experiments. This is because the administered dose can differ substantially from the delivered dose that reaches the cells. For example, for particles that settle rapidly, the difference between administering 100 µL per well or 200 µL per well of the same concentration will mean a doubling of the amount of material per well, and thus a potential doubling of the delivered dose. Moreover, since sonication influences the agglomeration state, and the agglomeration state influences the settling rate and thus the dose delivered to cells, determining the delivered dose may help the comparison of data among independent experiments using different dispersion protocols. A visual representation of two example NMs with different settling rates is shown in Figure 3. For SbD hazard testing specifically, determination of delivered dose aids in a comparison to benchmark materials with known toxicity, as settling may differ greatly between NMs. The importance of determining the delivered dose was shown in a study by Pal et al. (2015), where a correction for the delivered dose led to a considerable change in the hazard ranking of a panel of NMs, after which the in vitro outcomes matched better with the in vivo results [59].

The delivered dose can be modelled using the DG [60] or ISD3 model [61], which are currently available in the DosiGUI software generated in the PATROLS project [62]. The DosiGUI is user friendly, however, these models also introduce uncertainty as they do not take into account some critical factors such as particle convection [63], or dispersion-stabilizing surface functionalization. Moreover, cell stickiness needs to be chosen from an arbitrary scale, which is often an unknown parameter that has a big effect on the modelled delivered dose [64].

These models require the effective density of the NM as input, which is the density of the NM in a dispersion. In the case of agglomerates, this includes the density of the medium trapped inside the agglomerate. The effective density can be measured using analytical ultracentrifugation (AUC) or using the volumetric centrifugation method (VCM). In order to adhere to the criteria of SbD hazard testing, the VCM is preferred over AUC, as it is easier, less costly, and does not require specialized equipment [65]. It is important to measure model input parameters precisely, as small differences in input as a result of instrument variation may lead to large differences in the modelled deposited dose [64]. Stable dispersions are a requirement for modelling the dose rate and final dose delivered to the cells, as the calculations are based on a one-size distribution. The accuracy of the model outcome—and thus the estimated deposited dose (rate)—is less accurate if the size distribution of the dispersion changes over time due to agglomeration or aggregation.

The need to determine the delivered dose adds an extra step to SbD hazard testing, leading to a reduction of the achievable simplicity. Determining the delivered dose is however a requirement, even for SbD hazard testing, as more precise dosimetry will allow for more informative hazard testing. This, however, only applies to submerged testing, and not to experiments in which, for instance, an air-liquid interface (ALI) exposure protocol is followed. For ALI exposures, a quartz crystal microbalance (QCM) may provide sensitive and accurate deposited-dose measurements [66].

### 3.4. SbD Hazard Testing of NEPs and NMs Released during the Life Cycle

One of the most important aspects of SbD is assessing the safety of a product along its entire life cycle (LC) [5]. Usually, only pristine NMs are included in toxicity assays. This could be insufficient, as humans are also exposed to NEPs, aged NM and/or NMs released during the product LC including production, use, and end-of-life. The physicochemical characteristics of the NMs released to the environment along the different stages of the LC can be very different in terms of shape, chemical composition, agglomeration state, and surface modification [67,68,69,70,71]. Moreover, NEPs and NMs released during the LC might pose a different hazard than pristine NMs [72,73,74,75]. Thus, gathering information on the characteristics and hazard of NEPs and NMs released during the LC is important for designing relevant SbD interventions.

Processes leading to the release of NMs from a NEP during the entire LC can be simulated under laboratory conditions, after which NMs can be collected (e.g., by using filters) and redispersed in liquid for toxicity testing [76] or collected in liquid suspensions directly [77]. Realistic, released NMs, relevant for consumer exposure during the use of NEPs, can be obtained by using standardized methods (e.g., abrasion and weathering) that are normally used to test the durability and performance of NEPs. In the case of abrasion processes, there are different instruments that can be used to simulate mild or hard abrasion. Experimental parameters (e.g., cycles of abrasion, abrasion materials, normal load at the top of the abrader, etc.) can be tuned to reflect closely the NEP use conditions.

Aging experiments simulate conditions to which a product could be exposed during its use phase and are usually performed in a weathering chamber under accelerated conditions of UV exposure and rain. The weathering conditions (e.g., duration of cycles of light and rain, duration of the experiment, etc.) can be selected to follow international standards or be customized. In order to obtain higher quantities of released material (worst-case scenario), NEPs can be fragmented and sieved [78].

Released aerosols can be size-separated by using e.g., a cascade impactor to ensure inhalable or respirable fractions of NMs; After which they are collected on filters [79]. Efforts should be made to ensure high extraction efficiency and minimal compositional alterations when extracting material from filters [80]. Another option which is less easy but mimics better a real-life exposure is the direct exposure of cells to the released material, as performed by Zarcone and colleagues [81] for diesel exhaust. This approach is however somewhat more labour-intensive for SbD hazard testing but might be useful for gaining a more fundamental insight into the toxicity of released materials (exposure-relevant material) without losing a fraction to filter extraction.

After obtaining and extracting the material from filters, the same actions should be taken as for pristine NM testing, such as an accurate dose determination, controlling for interference, endotoxin contamination testing, choosing an appropriate dispersion protocol, etc. For NEPs and NMs released during the LC, endotoxin contamination might pose an extra challenge as these materials are generally not produced in sterile environments. Finally, compatibility in submerged settings might pose additional challenges as a NEP matrix is often plastic-based and might float on culture medium.

#### Feasibility and Relevance

Obtaining sufficient amounts of NM that are released at a given stage of the LC can be challenging due to low emission rates, contamination with other substances (e.g., sanding material), as well as laboursome and time-consuming procedures. The question is whether performing SbD hazard testing on pristine NMs is sufficiently relevant when assessing the safety of a NM along its entire life cycle. There are some examples in literature in which both pristine and released NMs have been tested to assess and compare their hazards. In most cases, materials released during the LC induced less or equal toxicity as compared to the pristine NM, as has been shown in in vivo studies [75,82,83] as well as in vitro [84]. This means that testing pristine materials, albeit often far from representing the reality, can still represent a worst-case scenario. In this case, risk screening of NMs released from a NEP can be mainly based on emission rates combined with the hazard information of the pristine NM. However, it should be noted that there is very little known about the toxicity of released NMs as compared to pristine NMs, and the exception may prove the rule.

Similarly, testing pristine NMs may represent a worst-case scenario for aged NMs. For example, freshly ground quartz particles have been shown to induce higher levels of pulmonary inflammation and cytotoxicity as compared to aged quartz [85].

For now, it should be considered on a case-by-case basis whether testing forms other than the pristine NM is required. More research is needed to determine whether the use of the pristine NM in SbD hazard testing is sufficient due to its ‘worst-case’ nature, or whether testing aged NMs, and NMs released during LC is crucial for designing SbD interventions.

It should be considered on a case-by-case basis whether testing forms other than the pristine NM is required. More research is needed to determine whether the use of the pristine NM in SbD hazard testing is sufficient due to its ‘worst-case’ nature, or whether testing aged NMs, and NMs released during LC is crucial for designing SbD interventions.

### 3.5. Challenging NMs and Advanced Materials

Hydrophobic NMs, NMs with low material density, multi-component NMs, and other advanced materials yet to be invented may show poor compatibility with commonly used in vitro assays. Applying a single standardized exposure method to all types of NMs will inherently give biased outcomes. For SbD hazard testing it is therefore important to consider the compatibility of NMs with challenging physicochemical properties and to be prepared for future novel advanced materials.

#### 3.5.1. Hydrophobic Particles

Since cells are always cultured in aqueous culture medium, hydrophobic particles can be of extra difficulty to test. Carbon nanotubes (CNTs) and graphene-based particles, for example, are notorious for being difficult to disperse in culture medium. Ethanol pre-wetting, using different dispersion media [86], and adjusting sonication time and frequency have been shown to improve dispersibility of NMs. However, for some NMs a stable dispersion can never be achieved in cell-culture medium. For example, some CNTs are specifically designed to agglomerate in order to reduce their dustiness and thereby improve their safety. For cases such as these, dry exposures at the air-liquid interface (ALI) should be considered when focussing on potentially respirable NMs. This requires a cell type that can be cultured on membranes in medium on the basal side, while being exposed to the air on the apical side. For the generation of a dry (dust) aerosol for ALI exposure, several methods are available [87]. However, it should be kept in mind that if a dust cannot be generated in a laboratory setting, inhalation of the NM is very unlikely. Thus, in these cases the relevance of an inhalation study should be reconsidered.

#### 3.5.2. Buoyant NMs

NMs with a density lower than cell-culture medium (e.g., certain types of plastic particles or agglomerates with a low effective density) will float and do not settle over time, resulting in no contact with adhering cells in a classical, submerged in vitro setup. This will likely lead to an underestimation of the potency of NMs such as nano-plastics and liposomes [88]. A solution to solve the problem of buoyant particles is to perform an inverted ‘overhead’ cell culture, where the cells are not cultured on the bottom of a culture dish, but upside down on top of the exposure medium. With this approach, it was possible to produce a dose response for several floating particles, whereas the traditional approach did not show any results [88,89]. For buoyant NMs, where inhalation is the relevant exposure route, ALI exposures can also be an option.

#### 3.5.3. Multicomponent NMs and Other Advanced Materials

In the past years, the more complex multicomponent nanomaterials (MCNM) have gained popularity. These next generation NMs consist of two or more materials or substances, giving rise to properties (e.g., reactivity) that are not equal to the sum of the properties of each component [90]. There are still many knowledge gaps when it comes to the toxicity of these and other novel advanced materials, which is why the concept of SbD is a suitable prevention-oriented approach. Whether these materials are compatible with the available toxicity assays is unknown, and this might pose challenges for future SbD hazard testing. Theoretically, MCNMs could exhibit multiple types of assay interference, attributed to the individual components of the MCNM. It is important to be aware of these challenges and to always assess interference.

## 4. Evaluation of In Vitro Methods for SbD Hazard Testing

### 4.1. Cytotoxicity

Measuring cell viability or cytotoxicity is a fundamental part of most hazard assessment strategies and integrated approaches to testing and assessment (IATA’s) for several reasons. Firstly, cytotoxic potency (for example LC_50_) gives an indication of the relative hazard of a NM. Secondly, cytotoxicity assays allow for the selection of appropriate sub-lethal doses for further mechanistic testing (e.g., genotoxicity and inflammation). Lastly, for several mechanistic assays such as genotoxicity assays, cytotoxicity measurements are a requirement for the correct interpretation of the results. Cell viability can be determined by the measurement of various cellular parameters, such as mitochondrial activity, lysosomal integrity, and membrane integrity. Different endpoints should be included to assess cytotoxicity [91,92,93], as results from different assays do not always correspond [94].

#### 4.1.1. Most Frequently Used Assays, Strengths and Limitations

The most-used approaches for measuring cytotoxicity or cell viability in vitro include measuring mitochondrial activity (examples are MTT, MTS, XTT, and WST-1 assays), release of cytoplasm components (examples are LDH and AK), lysosomal integrity (Neutral red uptake), apoptosis markers (caspase 3/7), and stains that can specifically enter apoptotic and/or necrotic cells (Trypan blue, Propidium iodide, and Annexin V). Propidium iodide and Annexin V can be combined to determine plasma membrane restructuring which can be representative of either necrosis or apoptosis specifically. Most cytotoxicity assays are relatively simple, can be carried out in a 96-well microplate format, could be used for HTS, and have commercial kits available. An exception are the assays that require microscopic evaluation of a certain staining, as this is more labour-intensive.

For many cytotoxicity and viability assays, NMs can interfere with assay reagents and/or the optical readout [95,96,97,98]. Therefore, potential interference of the NMs with assays should always be assessed [92,99]. The elimination of NMs via high-speed centrifugation may reduce optical interference [92]. Alternatively, since mitochondrial activity is measured intracellularly, NMs can be washed away from cells prior to incubation with the reagent to avoid interaction of the NMs with the reagent [100]. For products measured in the supernatant (e.g., LDH and AK), washing is not feasible, but centrifugation can help remove larger NMs and thereby reduce optical interference. However, some NMs are known to inactivate or adsorb LDH directly [101]. If interference still occurs after taking precautions, it is advised to perform another type of cytotoxicity test, as the subtraction of the average background signal of the NMs will reduce the accuracy of the outcome [97,102,103]. For specific NMs, some assays might prove not to be compatible.

Much effort has been put into the optimization and standardization of in vitro cytotoxicity assays specifically for NMs in the past years. An ISO standard for the MTS assay was published in 2018 [104]. In 2021, an ISO standard was published for impedance measurements for NMs specifically [105]. This assay involves growing cells on an electrode during exposure to the NM. The detachment of cells, indicating cytotoxicity, is measured as a decrease in electrochemical impedance, as demonstrated in the assessment of poly-lactic acid NM-induced toxicity in A549 epithelial cells [106]. This assay is less prone to interferences as no optical readout and no assay reagents are required. However, it does require specialized equipment not available in many labs. Internationally standardized and harmonized standard operating procedures (SOPs) for other cytotoxicity assays have not been published to date.

In the NanoReg project, an interlaboratory study for the MTS assay was carried out, and acceptable robustness levels were found depending on the cell type. The human alveolar cell-line A549 showed a good agreement in cytotoxicity between labs, whereas the differentiated human monocyte cell-line THP-1 (dTHP-1) showed varying results and a poor robustness [107]. In a large interlaboratory study by Piret et al. (2017), a good robustness was found for the MTS assay and ATP content measurements. These comparisons were carried out using both A549 as well as dTHP-1 cell lines, and two different NMs. The authors stressed the importance of avoiding interference of the NM with the assay in order to obtain more reliable results, and a lower inter-laboratory variability. They also found that the caspase-3/7 assay showed a high inter-laboratory variability [100].

A large interlaboratory study of eight labs studied how to improve the robustness of the LDH and MTS assay. After a first round of experiments, adaptations to the protocols were made and robustness increased significantly within and between laboratories. Changes made to the protocols included the optimization of the differentiation of THP-1 cells and centrifugation after incubation with MTS reagent to remove NMs [108]. These findings on the MTS assay were confirmed in another interlaboratory study using the A549 cell line. Additional sources of variability were identified in this study. A549 cells from two different suppliers showed a large difference in cytotoxicity in response to polystyrene NMs. Also, the inclusion of serum effectuated large differences in cytotoxicity as compared with serum-free experiments. Moreover, differences in pipetting techniques (e.g., harsh aspiration vs. gentle pipetting and completely removing medium vs. partially removing medium before MTS incubation) and dispersion protocols were identified as causing differences in results between laboratories [52]. The importance of more elaborate and detailed SOPs was again stressed in a recent inter-laboratory study, where the inclusion of several acceptance criteria was found to improve the robustness of the MTS assay, such as maximum acceptable variations between replicates, minimum cell survival, and maximum interference levels [102].

#### 4.1.2. Predictivity and Relevance

Whether in vitro cytotoxicity assays are predictive of in vivo acute toxicity has been studied for years for soluble chemicals. For NMs, however, there are only a few studies that correlate in vitro cytotoxicity with in vivo toxicity. Therefore, in this section we have included not only studies that correlate in vitro cytotoxicity with in vivo markers of cell death (apoptosis, necrosis etc.), but also with any type of in vivo toxicity.

In general, the predictivity of cytotoxicity assays depends on the mechanism of action of the NM, as well as on the cell type used for the in vitro study [109,110,111]. NMs which exert their effect through the shedding of toxic ions are usually also cytotoxic in vitro [100]. In a comprehensive comparison study, in vitro cytotoxicity was compared to in vivo lung inflammation for several different particles, using comparable doses. LDH release and trypan blue exclusion assays were able to predict the inflammation-inducing effects of ion-shedding NMs, but not of poorly soluble NMs [112]. However, in another study, in vitro LDH release in response to poorly soluble TiO_2_ NMs correlated well to the in vivo number of polymorphonuclear cells (PMN) in BALF. This correlation was only present when the dose was expressed as surface area, and not when using mass as dose metric [113].

The toxic effects of CNTs in vivo upon inhalation are not easily predicted using in vitro cytotoxicity assays, unless the toxicity is caused by metal impurities [109]. Also for CNTs, the in vitro effects differ between cell types [114]. Other carbon-based materials, such as diesel exhaust also do not show an accurate correlation of cytotoxic response with in vivo effects. LDH release from A549 cells and LDH measured in BALF from rats upon instillation with diesel exhaust did not correspond, and even showed an opposite ranking in toxicity [115]. However, the suspensions used were not purely the particle fraction and contained other substances such as lube oil (which floats on culture medium), possibly causing the contrasting rankings.

The choice of cell type is crucial for performing a predictive in vitro cytotoxicity assay. For example, the WST-1 and NRU assays were able to establish an accurate ranking in toxicity of Ag, Au, SiO_2,_ and MWCNTs, but IC50 values differed between the cell types used [114]. In a study by Sayes and colleagues, in vitro LDH release did not correlate with rat pulmonary LDH release and inflammation (% PMNs) for rat primary pulmonary macrophages and rat pulmonary epithelial L2 cells grown in mono-cultures. However, when grown in co-culture, in vivo LDH release and inflammation were accurately predicted via the in vitro LDH release for crystalline silica and ZnO (but not for amorphous silica) [94]. A similar study also showed a good correlation for this co-culture model for ZnO NM, but only at the highest (particle overload) dose [116].

When choosing a cell line, immune cells are found to give a higher prediction accuracy than fibroblasts [114]. When macrophages are thought to be involved in the toxicity of NM, it is especially important to select an immune cell type for testing cytotoxicity. THP-1 cells which were differentiated to macrophages (dTHP-1) showed a higher sensitivity for cytotoxic effects as compared to A549 cells (alveolar cell line) for a panel of 24 NMs [117]. Cho and colleagues found that differentiated peripheral blood mononuclear cells and isolated lung macrophages performed better compared to cell lines such as dTHP-1, A549, and 16-HBE [112]. The fact that primary cells are more sensitive than cell lines is generally accepted. Despite this, primary cells are more difficult to work with and more expensive, and will therefore most likely be disfavoured for SbD hazard testing.

#### 4.1.3. Overview of Needs and Knowledge Gaps

Table 1 shows a summary of how the different cytotoxicity assays perform in terms of the criteria for SbD hazard testing. As cytotoxicity measurements are a requirement for several mechanistic assays, they are crucial for SbD hazard testing. Simple cytotoxicity assays—although optimizations for NMs are needed—serve as a good starting point for detecting hazard warnings in SbD hazard testing. It is recommended to include at least two different cytotoxicity assays as different assays measure different mechanisms [92,118]. A combination of a mitochondrial activity assay and a membrane-integrity assay is recommended.

Prediction accuracy of cytotoxicity assays should be investigated further. It was shown that predictivity depends on the cell type used and the mode of action (MOA) of the NM. Assay applicability domains should be mapped in more detail to understand which toxic effects can be predicted with in vitro cytotoxicity assays and which cannot. We also found that protocol optimization improves assay robustness. Moreover, interferences are quite common for cytotoxicity assays, and they can be avoided by taking the right precautions and including the right controls, which is crucial even when performing SbD hazard testing. Together, this indicates the need for optimised and standardized protocols for NMs specifically. This will in turn also aid the determination of the prediction accuracy of assays.

As cytotoxicity measurements are a requirement for several mechanistic assays, they are crucial for SbD hazard testing. Simple cytotoxicity assays—although optimizations for NMs are needed—serve as a good starting point for detecting hazard warnings in SbD hazard testing. It is recommended to include at least two different cytotoxicity assays as different assays measure different mechanisms A combination of a mitochondrial activity assay and a membrane integrity assay is recommended.

### 4.2. Dissolution

Although in vitro testing of dissolution is a measure of a PC property, and not directly a measure of toxicity, the results obtained can be used to infer potential toxicity, or even potential pathogenicity. This is through consideration of a material’s biodurability or its transformation to ions or molecules. In this context, biodurability may be accompanied with biopersistence, which historically has been linked to the fibre pathogenicity paradigm such as that relating to asbestos, CNTs, and other respirable fibres [119], but also in relation to poorly soluble particles such as TiO_2_. Long-term inhalation exposure to poorly soluble particles can induce impaired clearance and chronic inflammation that might even progress to cancer, as has been observed for TiO_2_ in rats [120]. The human relevance of these results is a topic currently receiving a resurgence in interest within the scientific community [121]. Conversely, rapid dissolution of a substance can indicate exposure to potentially harmful soluble components, such as metal ions, which can be released in body compartments that are otherwise inaccessible.

Information on dissolution in relevant conditions is greatly beneficial for the hazard assessment of NMs and in defining SbD interventions. Information on dissolution is already a requirement of REACH and EFSA [21] and dissolution rates are a valuable criterion within all of the current risk assessment tools available for NM hazard assessment and can also be used for grouping/read-across [14,16,18].

#### 4.2.1. Most Frequently Used Assays, Strengths and Limitations

There are various methods used in in vitro testing of dissolution, acellular and cellular, which have not changed significantly for some time. There are a number of guidance documents including ones from ISO (ISO 19057:2017) and OECD (OECD GD No. 318, specifically for environmental studies) which provide the start of standardization of these techniques. The output of various EU projects (e.g., GRACIOUS, Gov4Nano and BIORIMA) will also greatly impact the development of this methodology.

For acellular testing, it is possible to test within static systems [122] or flow-through (dynamic) systems [123,124]. The application of these methods is extremely diverse, as the formulation of different simulant fluids may facilitate the simulation of any biological compartment, including extracellular and intracellular compartments, and any exposure route of interest including oral, dermal, or inhalation [125], with recommendations made within an ISO technical report (ISO 19057:2017). There are a number of differences, both subtle and substantial, in the simulant fluids used that determine the accuracy of in vivo prediction. For example, components such as citric acid have a significant effect on the dissolution of certain metals, and inclusion of proteins/serums will also have an effect on dissolution [126]. These considerations have been recently reviewed [127]. Two recently completed projects (nanoGRAVUR and GRACIOUS) identify the abiotic flow-through system ISO/TR 19057:2017 as the most relevant system [124,128], with a technology readiness level (TRL) identified as high/medium for metals using Inductively Coupled Plasma Mass Spectroscopy (ICP-MS) analysis and medium/low for materials such as CNTs that require techniques such as Transmission Electron Microscopy (TEM) or X-Ray Photoelectron Spectroscopy (XPS).

Although there are currently no accepted methods for in vitro cellular dissolution testing, various studies have been conducted and these may be more reflective of the in vivo response following inhalation of particles, although there have been concerns raised with the cellular methods.

##### Acellular Methods

Acellular dissolution can certainly be considered simple, considering the practical requirements. However, each methodology has different demands and associated limitations. For example, the solutes released during dissolution within a static system, especially those of a basic nature, may cause enhanced nucleation, precipitation, changes in localised pH, and/or saturation effects preventing further dissolution [129,130,131,132]; this behaviour is unlikely to reflect the in vivo behaviour, demonstrating clear limitations of static systems. Although dynamic systems, by design, circumvent these issues, they are not without limitations, and there are a number of factors which may affect their reproducibility [132]. NMs may pass through filters used in flow-through systems, leading to misinterpretation of results and potential false-positive results [133], or filters may become blocked and ruptured by components of the more complex fluids such as proteins or lipids [134]. Practically, dynamic systems are cumbersome due to the high volume of liquids required in long-running tests [135]. Although a comparison of acellular methods is not often made, when done so the findings have been confounding. For example, the dissolution rate of gold nanoparticles has been found similar in static and dynamic systems [136], while the solubility of BaSO_4_ has been shown to differ in static and dynamic systems [137].

It is often reported that distinction between different material forms is possible, with a level of sensitivity allowing for a distinction between dissolution rates leading to grouping, as has been suggested for fibrous NMs within the GRACIOUS project [138]. This approach has been aligned with previously established methodology for man-made vitreous fibres (MMVF) and asbestos, whereby NMs biodurability can be defined by the respective dissolution in alveolar fluid and lysosomal fluid. Good comparability has already been found for in vitro dissolution of MMVF materials and in vivo biopersistence [139], allowing confidence in this approach. Heavy influencers of sensitivity include the analytical method used to detect released ions, as well as high background measurements caused by the complexity of fluids used, although this can be alleviated through the removal (or reduction) of specific metal components within the fluid, in line with the solutes expected to be released from the test material [140]. The current use of dissolution within RA tools may not be so greatly impacted by sensitivity, as the thresholds used are very broad and/or rather elementary, using a ranking based on dissolution time (ANSES, Swiss Precautionary Matrix) or soluble concentration (GUIDEnano). Advances have been made recently to include threshold decisions based on dissolution rate [21,123]. Nevertheless, unless very significant changes are made to the particle to result in very different dissolution behaviour, it is unlikely that the sensitivity of the thresholds used will be dynamic enough to provide meaningful SbD decisions on dissolution behaviour.

In terms of compatibility, the potential of acellular tests is broad, and other than particular hydrophobic materials, it is difficult to list examples that could not be tested. In fact, these acellular methods have been used for some time to resolve the time-kinetic release of metals within complex materials, such as man-made fibres, or from occupational dusts such as welding fumes [141]. The biological predictivity of acellular tests, although not always established within the literature, has been demonstrated to a relatively high level, with various promising outcomes. For example, the solubility of BaSO_4_ in the dynamic system was considered to reproduce what is known for the solubility of BaSO_4_ in vivo, while the results of the static system underestimated this [137]. With the use of a lysosomal simulant fluid, the dynamic dissolution system was also shown to replicate cellular dissolution of BaSO_4_ (in conjunction with SrCO_3_ and ZnO) in rat macrophage models [123], and similarly acellular dissolution of MoO_3_ in the same lysosomal simulant fluid was found comparable to dissolution within mouse macrophage models [142]. These studies, and others, have demonstrated that by using various simulated biological fluids of intracellular compartments and/or lung lining fluid, a number of correlations with either cellular assays or in vivo exposures can be attained [123,143,144,145,146]; however, it should be acknowledged that there is an equal number of studies that have shown no correlation, raising the concern for appropriate fluid selection [127].

##### Cellular Methods

The basic principle of the cellular method is simple and can be performed cheaply as typical assessments investigate dissolution within cells up to 24 h. The difficulty with this methodology is the success of analysing the ions released. There are various options for separating cells and supernatant, such as centrifugal ultrafiltration and cloud point extraction [147], however it is not always as straightforward as the acellular assessments, as released ions may form complexes with biomolecules and therefore separation may be hampered [148]. Additional concerns may arise from the complexing of ions to biomolecules. Therefore, studies have often opted to determine both the ion concentration and the NM concentration to aim to avoid false positives or false negatives [149].

Koltermann-Jülly et al. (2018) found that macrophage-assisted dissolution in vitro was only applicable for an exposure period of 1–2 days, which they believe is too short and may be responsible for the low amounts of ions detected for the NMs tested [123]. The authors state that using cellular systems gives no additional benefit to the abiotic flow-through system with regards to predicting the in vivo response. This conclusion, however, is based only on the three materials tested. Moreover, when a cellular method uses uptake as an inference of dissolution, overestimating particle concentrations may occur, when measurements are not only of internalised particles but also of those adhered to the cell membrane. However, this is likely to be resolved by following well-described methodology which includes steps to limit this interference such as ensuring thorough washing and etching of the cells prior to analysis to remove adhered particles [150]. Further promising methodologies for this include the isolation of NMs and ions from cells using Triton X-114-based cloud point extraction as has previously been conducted for intracellular Ag NMs and Ag+ isolation [147,149].

#### 4.2.2. Overview of Needs and Knowledge Gaps

Table 2 shows a summary of how the different dissolution assays perform in terms of the criteria for SbD hazard testing. There is a wealth of studies available for interpretation of in vitro dissolution methods, and although there are promising findings, there are still too many uncertainties to be sure of which model is most appropriate or reliable for specific materials. For SbD hazard testing, the use of a static system would be preferred, due to its simplicity. Although correlations with in vivo outcomes have been shown for static and flow-through systems, this is not always the case, and therefore requires further attention. Going forward, assessing predictivity will be important, especially when assessing novel materials; however, auspiciously, as shown above, for some substances a strong relationship between acellular, cellular, and in vivo findings has already been observed. It has been previously suggested that specific in vitro methods (e.g., specific fluid choices) should be selected based on feasible degradation pathways [142] which could be dependent upon specific degradation routes e.g., complexation, protonation, or to establish robust and fully accurate biological simulations [127]. Moreover, if these methods are to be applied in grouping and read-across approaches, the development of reliable and robust methods for determining particle dissolution rates has been considered paramount [123].

There is a wealth of studies available for interpretation of in vitro dissolution methods, and although there are promising findings, there are still too many uncertainties to be sure of which model is most appropriate or reliable for specific materials. For SbD hazard testing, the use of a static system would be preferred, due to its simplicity.

### 4.3. Oxidative Potential and Oxidative Stress

The oxidative potential (OP) of a NM is a chemical property that defines the ability of a NM to form potentially toxic species such as hydroxyl (^•^OH) and superoxide (O_2_^−^) radicals and hydrogen peroxide (H_2_O_2_) (collectively called reactive oxygen species (ROS)), or reactive nitrogen species (RNS) through redox reactions. This parameter is part of many NM hazard assessment strategies and grouping approaches due to its potential as a predictor of toxicity [13,14,15,18]. The pros and cons of OP assays in NM research have extensively been reviewed previously [156,157].

Oxidative stress (OS) is a cellular state in which the amount of ROS, caused by NM OP or the release of reactive ions, overwhelms the cells’ antioxidant capacity, potentially leading to the oxidation of biomolecules, inflammation, and oxidative DNA damage [158,159]. OS is seen as an important key event in the mode of action of many NMs and is therefore important to quantify as an early warning indictor [156,160,161].

#### 4.3.1. Most Frequently Used Assays, Strengths and Limitations

##### Acellular Methods

In an acellular assay, OP is usually measured as a rate of depletion of a reductor. OP assays do not measure reductor depletion by OP only, since the release of reactive ions by dissolution will also lead to a depletion. Multiple acellular assays have been proposed to evaluate NM OP. The acellular dichloro-dihydro-fluorescein (DCFH) assay, electron paramagnetic resonance (EPR), electron spin resonance (ESR) spectroscopy, and the Ferric Reducing Ability of Serum (FRAS) assay are frequently used and have been evaluated extensively in literature [162,163,164,165,166,167]. An additional assay which is especially relevant for measuring NM OP is the haemolysis assay. For this assay, red blood cells are isolated from whole blood, and the ability of NMs to disturb their membranes is measured through absorbance. This assay requires whole blood, but no cell culturing, making it an easy and cost-effective method. No interferences have been reported in this assay, but it has not been studied extensively for NMs. There is no information available on robustness, and there is no publicly available standardized protocol.

Out of the other acellular assays, standardized protocols are only available for EPR/ESR (i.e., ISO 18827:2017). These assays require relatively expensive equipment, not always available in standard laboratories. FRAS and DCFH are considered simple and low-cost assays, requiring equipment that is present in any standard biological lab [165,166,168]. The FRAS method was originally developed to measure the ferric reduction in blood plasma (FRAP) [169]. It has been adapted to be used with serum, optimized for smaller volumes [165] and for multi-dose measurements, while showing good sensitivity and reproducibility for several metal-bearing NMs [163]. Recently, the FRAS protocol was successfully adapted to measure the reactivity of graphene-based materials by adding a filtration step for NMs of very low density [168]. Interlaboratory studies for the FRAS assay have not yet been performed.

The robustness of the acellular DCFH assay using different NMs was evaluated in a recent inter-laboratory study. A good robustness was found for the positive control NMs when normalizing fluorescence values between labs. However, for the other NMs, interlaboratory reproducibility differed per particle type [170]. Several papers reported NM interference with, for example, the fluorescent readout of the DCFH assay [162,166,171]. Zhao and Riediker (2014) identified several other factors that could reduce the reliability of the acellular DCFH assay, such as the use of different dispersing agents, as well as using too-high concentrations of NM [172]. Interference by way of NM flocculation or optical interference has been noted regarding the FRAS assay when testing various NM pigments [173], while no NM interference has been reported for ESR/EPR.

The EPR, DCFH, and FRAS assays have recently been evaluated for NM-grouping purposes. Results showed that the sensitivity of the methods greatly depends on the type of particle studied. For example, CuO, BaSO_4_, and Mn_2_O_3_ were consistent in their reactivity level across the three methods, but ZnO and CeO_2_ only showed a response in the FRAS assay, and not in EPR and DCFH measurements [173]. This might suggest that reactive species produced by certain NMs are captured better by some assays than by others, or that the FRAS assay is more sensitive in general. The latter has been confirmed in several studies that showed that the FRAS and ESR/EPR perform better than the DCFH assay in terms of sensitivity [163,171,174,175].

The choice of assay should depend on the goal of testing. For example, if one would like to know which types of radicals are formed in order to know what to change in the NM design as a SbD intervention, ESR/EPR measurements with different spin traps will provide the most informative results [176]. The FRAS assay can provide a more general image of ROS generating potential, as a result of the cocktail of antioxidants that is present in serum. The DCFH assay is especially sensitive for one-electron oxidizing species (such as hydroxyl radicals) [177]. It should be taken into account that these assays measure the OP of the NM in the specific environment required by the assay. OP is greatly influenced by the exposure environment, and therefore the OP measured in the assay may not fully reflect the OP in a real-life exposure scenario. A relevant protein corona could be applied to ensure exposure relevancy, as is described in Section 3.2.

##### Cellular Methods

Cell-based assays can directly measure the intracellular ROS, irrespective of their origin (i.e., as a result of the surface chemistry of the NM, as a cell-generated signalling molecule, or as a defence mechanism of the cell within the phagolysosome), for example through use of the cellular DCFH-DA assay. Other options include the assessment of the effect of these radicals on biomolecules such as lipids (e.g., lipid peroxidation) and proteins (e.g., protein carbonylation), cellular antioxidant status (e.g., glutathione (GSH:GSSG ratio)), and antioxidant gene regulation (HO-1 expression and Nrf-2 reporter cell lines), of which the latter two are extensively described in Boyles et al. (2016) [178].

It has been suggested that OS measured in a cell-system has advantages over measuring the OP in acellular systems. By measuring in a cellular environment, the cells’ ability to defend itself against the induced OS is taken into account, the ROS’ generated genotoxicity can be assessed, and other mechanisms other than the OP that lead to OS are captured as well [156]. Other mechanisms leading to OS, such as through mitochondrial perturbation, have been shown for chemicals extracted from diesel exhaust particles [179,180], yet there is no convincing evidence that NMs are capable of inducing OS through mechanisms other than OP or ion release.

#### 4.3.2. Predictivity and Relevance

For SbD hazard testing, it would be desirable to be able to predict human health effects or at least effects that are observed in studies in experimental animals with simple, fast, and cheap acellular OP assays. The ability of acellular assays to predict cellular oxidative stress and in vivo oxidative stress markers is quite good, as shown in a comprehensive review by Moller et al. (2010), but not for all particles and all test systems [160]. It has been shown that NMs can induce different types of ROS [181] and therefore it depends on the type of NM and their MOA whether assays can predict cellular and in vivo effects. For example, data derived with the haemolysis assay correlate very well with in vivo pulmonary inflammation for a panel of 13 metal oxide NMs (92% prediction accuracy), whereas EPR (69% prediction accuracy) and DCFH (77% prediction accuracy) results showed lower correlations [164]. The haemolysis assay was able to predict in vivo pro-inflammatory responses of both NMs that act through soluble ions as well as NMs that act through surface reactivity with a prediction accuracy of 62.5% for a panel of eight NMs [112]. The FRAS assay has even been shown to be able to correctly distinguish OPs between several types of CNTs [175]. In a study comparing pulmonary inflammation (PMN influxes) upon inhalation of a range of NMs with acellular ESR and DCFH results, the correlation was reasonable; however, here it was concluded that ESR measurements in macrophages give a higher prediction accuracy than the acellular assays [182]. For SiO_2_ NMs, EPR results correlated very well with in vitro cellular cytotoxicity [183]. ESR also correlated well with in vitro protein carbonylation for a large panel of NMs [184]. However, ESR as well as the FRAS assay were able to accurately predict only 50% of the in vivo outcomes for a panel of 35 NMs [162].

False positives in acellular OP assays (when compared to in vivo outcomes) can be explained by the fact that cells and organisms can resolve ROS to a certain extent. Therefore, effort should go towards establishing thresholds for these assays. False negatives in acellular OP assays can be explained by the fact that other mechanisms other than OP can lead to pulmonary inflammation as well, which cannot be detected by these assays. The large variation between prediction accuracies between studies could be explained by the differences between the NM panels tested. Each assay has a specific applicability domain and prediction accuracy will therefore depend on the NM types and the resulting types of ROS.

In general, cellular assays show a higher prediction accuracy than acellular assays, and a combination of both might perform even better [157,162,185]. However, for SbD hazard testing, acellular assays may already give a good indication of toxicity and could serve as a valuable initial screening in the very early stages of NM development.

#### 4.3.3. Overview of Needs and Knowledge Gaps

Table 3 shows a summary of how the different OP assays perform in terms of the criteria for SbD hazard testing. There are clear indications that only measuring acellular reactivity would be sufficient for SbD hazard testing, when cellular testing is already performed for cytotoxicity, genotoxicity, and pro-inflammatory effects. OP assays can be used to categorize the materials and to explore in more depth if the OP can or should be reduced in a SbD intervention. Mapping the prediction accuracy for each assay, as well as an applicability domain will help understand which assays can be used to predict which specific effects.

There are clear indications that only measuring acellular reactivity would be sufficient for SbD hazard testing, when cellular testing is already performed for cytotoxicity, genotoxicity, and pro-inflammatory effects.

### 4.4. Inflammation

Many IATAs and testing strategies include the measurement of inflammatory potential using NMs since this is generally accepted as one of the key mechanisms of NM toxicity [15,19,20]. Pulmonary inflammation in response to NM exposure has been shown to lead to several adverse health effects, such as fibrosis as well as lung cancer in animal studies [120,186,187]. For oral exposure, inflammation is a key parameter in NM toxicity as well [188]. However, this section will focus on assays targeting the pulmonary route of exposure only.

#### 4.4.1. Most Frequently Used Assays, Strengths and Limitations

It is impossible to capture the complexity of an in vivo inflammatory response in an in vitro model, where recruitment of inflammatory cells other than those already present cannot occur. It is however possible to detect the cytokines responsible for this recruitment in an in vitro experiment. The most widely used approach to assess inflammatory responses in in vitro assays is measuring cytokine production or secretion, using, for instance, an enzyme-linked immunosorbent assay (ELISA), RT-qPCR, or multiplex-based immunoassays [189] after exposing cultured cells to NMs. Measuring the levels of pro-inflammatory cytokines and other inflammatory mediators may give insight into the mechanisms of the immunomodulatory effects of NMs in vitro, such as inflammasome activation or dendritic cell maturation. Cytokines of specific interest for NM pulmonary toxicity are, amongst others, IL-8 as markers for neutrophil recruitment [190], IL-1β as a marker for NLRP3 inflammasome activation [191], and TNF-α as a marker for macrophage activation [192].

Cytokine release can be measured in e.g., epithelial cells, macrophages, and dendritic cells, cultured in mono- and co-cultures. Cells can be exposed in a submerged setup or at the air-liquid interface (ALI), where cells are cultured in contact with the air and exposed to aerosols on the apical side whilst kept in medium on the basal side, better resembling the physiological environment of cells in the respiratory tract (Figure 4).

A critical factor when assessing pro-inflammatory responses in cell models is that some NMs can interfere with common in vitro assays. SWCNT and MWCNT can non-specifically adsorb TNF-α and IL-8 to their surface, and TiO_2_ NMs have been described to be able to adsorb IL-8, thus causing a false-negative result in ELISA assays [99,118]. This effect has also been observed for Ag NM in combination with TNF-α and IL-8 [100]. NMs are also known to interfere with the components of the ELISA. This problem can be overcome by centrifugation to remove the NMs from the supernatant before performing the ELISA. It is also essential to test the NMs for endotoxin contamination, as endotoxins can induce inflammation at very low concentrations, leading to false positive results [193,194], especially since NMs are generally not produced in a sterile environment.

Despite the relevance of pro-inflammatory effects of NMs in human health, there is currently no validated test method available to investigate inflammatory responses in vitro. Submerged assays have been used far longer compared to the relatively new ALI models, and thus more advances in standardization and optimization have been accomplished. Only a few studies have been performed to show the robustness of one of these protocols. At the ALI, Calu-3 cells with and without macrophages (either differentiated THP-1 cells (dTHP-1) or primary cells) showed high reproducibility in seven participating labs based on measurements of membrane integrity and mitochondrial activity. Cytokine release however showed higher variability, although similar trends between the seven labs were observed [195]. The reproducibility between labs after exposure of A549 cells at the ALI to lipopolysaccharide (LPS) as a positive control was found to be quite low. However, after protocol optimizations, special training of personnel for cell handling, and homogenization of disposables and reagents, the reproducibility increased [196]. The reproducibility of results between labs when using dTHP-1 cells is a frequent topic of debate. Not only do they show varying responses to NMs, but also to a positive control such as LPS, as shown in a large inter-laboratory study [100]. In another large inter-laboratory study by Xia et al. (2013), it was shown that good results can be obtained when using very detailed protocols and using the same batch of serum and cells. They also showed that cell-culture conditions and the duration of differentiation greatly affect the variability of dTHP-1 cells between labs [108].

A frequently used alternative for dTHP-1 macrophages are monocyte derived macrophages (MDMs), derived from donor blood. Even though they are considered more predictive of the in vivo situation, they are also known for their donor-to-donor variation. The same holds true for the use of commercially available primary epithelial cells, which are considered more relevant, but also show considerable variation [197].

In ALI exposure systems, there are many other factors that may contribute to an increased variability, such as the accuracy of the microbalance in the exposure system, the quality of the nebulizer used, the method of sample preparation, etc. A comprehensive overview of factors that can influence reliability and robustness can be found in Petersen et al. (2021) [198].

In terms of predictivity, commercially available primary cells generally give a good indication of in vivo effects for known inflammation-inducing particles. Studies have shown that the pro-inflammatory effects of quartz [197], Ag NMs [199], SiO_2_ NMs [200], and Pd and Cu NMs [201] are accurately predicted using primary cell models. Co-cultures of cell lines are also able to predict in vivo responses in many cases, as has been shown for quartz [94,202], CuO [112], and ZnO [94]. In this latter study, it was shown that co-cultures perform better than the two cell types separately, showing the importance of interplay between epithelial cells and immune cells. The addition of macrophages seems to be crucial in order to capture a much wider domain of immunological responses as compared to epithelial cells only, as has been shown in multiple studies [108,203,204]. Epithelial cell lines in mono-cultures were not able to predict the toxic effects of quartz [205], but did accurately detect Ag NMs’ pro-inflammatory effects [112].

In short, primary cells are the most sensitive, followed by co-culture systems with macrophages, and then mono-cultures. There are however some studies that prove otherwise. Cho et al. (2013) showed that cell lines performed similar to primary alveolar macrophages and differentiated PBMCs in terms of accuracy [112]. The A549 epithelial cell line in tri-culture with inflammatory cells did not pick up the pro-inflammatory effect of Ag NMs [206]. Mono-cultures of the epithelial cell line 16-HBE better predicted in vivo effects of Ag NMs than when in co-culture with dTHP-1 cells [207]. Furthermore, CeO_2_, Co_3_O_4_, and NiO NMs induced an increase in granulocytes in BALF, whereas no pro-inflammatory effects were seen in submerged mono- and co-cultures [112]. Finally, BEAS-2B and dTHP-1 cells were able to predict a ranking in pro-inflammatory effects of several types of CNTs which corresponded to in vivo markers of lung fibrosis in two separate studies [208,209]. This could mean that cell lines could be suitable for SbD hazard testing. Likely, different modes of action of toxicity require different levels of complexity in a cell model. In order to be sure about the predictive capacity of the different cell types, more types of NMs should be tested.

The exposure method chosen will also impact the predictivity of the method. Exposing ALI-cultured cells is generally considered a more sensitive approach, since it is more physiologically relevant, as has been shown in multiple studies [210,211,212]. However, for SbD hazard testing it is desirable to work with a model that is as simple and cost-effective as possible. This disfavours the use of primary cells and favours simple submerged exposure systems as opposed to the more complex ALI cultures. There are strong indications that simple submerged models could be predictive enough for SbD hazard testing. For example, in a study by Loret et al. (2016) they concluded that, indeed, co-cultures were more sensitive than monocultures, and that ALI exposures were more sensitive than submerged cultures. However, the general ranking of the NMs in terms of their toxicity was similar across the various exposure methods [213]. A study by Di Ianni et al. (2021) showed a strong correlation between in vitro submerged co-cultures and in vivo results when testing CNTs [190]. In a study by Herzog et al. (2014), the pro-inflammatory effects of Ag NMs were not detected in ALI culture conditions, but were detected under submerged conditions, suggesting a better performance of the submerged model [214]. Submerged and ALI exposures performed equally well for cytotoxicity in response to TiO_2_ [215]. In a study by Panas et al. (2014), submerged conditions were more sensitive in detecting the pro-inflammatory effects of SiO_2_ NMs as compared to ALI conditions [216]. Altogether, the potential for submerged experiments to predict in vivo responses has been shown in multiple studies, and is worth exploring further, especially for SbD hazard testing.

The differences in sensitivity between the ALI and submerged exposures can be explained by many (potentially confounding) factors. Firstly, certain cell types such as A549 are suggested to produce surfactant at the ALI, but not under submerged conditions, making them more vulnerable to toxic effects in a submerged experiment [217]. Secondly, the effective dose in submerged experiments is not always (correctly) calculated, and this may lead to a skewed comparison to ALI and in vivo results. Thirdly, the medium used in the submerged experiments may have an impact on NM behaviour in terms of the protein corona and dissolution rate, which does not occur, or occurs differently, in ALI experiments. Lastly, studies finding a good correlation between an in vitro model and in vivo results are more likely to be published, leading to publication bias. In a comprehensive overview of different cell types and exposure methods by McLean et al. (in preparation), it was shown that for quartz hazard prediction, the strength of in vitro prediction of in vivo responses was highly inconsistent, and largely dependent upon the data and study quality, which highlighted a need for robust SOPs which take into account numerous requirements for in vitro/in vivo extrapolation (McLean et al., in preparation).

There are several advantages of using ALI over submerged exposures. Particle alterations due to interaction with medium (such as the formation of a protein corona, dissolution, agglomeration) are no longer an issue, and calculating the deposited dose is much easier as compared to submerged experiments [212,218]. ALI exposures are compatible with a wider range of NMs, including hydrophobic particles, as exposures can be performed using a powder. Without having to take into consideration their behaviour in medium, abrasion products of NEPs can also directly be applied, making this type of model especially interesting for assessing life-cycle considerations, as is crucial for SbD hazard testing. Using ALI exposures is however more time-consuming and less high-throughput. Additionally, robustness of deposited dose after nebulization in an ALI setup may be low, depending on the NM used [219].

#### 4.4.2. Overview of Needs and Knowledge Gaps

Table 4 shows a summary of how the different inflammation assays perform in terms of the criteria for SbD hazard testing. For SbD hazard testing, it is crucial to include tests that are predictive yet simple and cost-effective. Therefore, based on the current literature, the use of a submerged co-culture model including at least a type of macrophage might be the most suitable. However, more research is needed to confirm that simple methods are predictive enough for early hazard screening by testing data-rich NMs. Moreover, novel and advanced NMs should be tested in the available cell models in order to determine the compatibility of the cell models and readouts with different types of NMs. For a better predictivity, avoiding issues with dosimetry and medium interactions, and for hydrophobic particles, a simple ALI experiment can be set-up for SbD hazard testing.

A short-lived inflammatory response is beneficial to help clear NMs from the lung, and macrophage recruitment may not necessarily be a hazard warning. There is still a poor understanding of which amount of inflammation could be considered an adverse outcome, especially when measured in vitro. Establishing meaningful thresholds for these assays is important. Moreover, since pulmonary inflammation is mostly a chronic adverse effect, more work should be focusing on predicting chronic effects with in vitro assays, with which a promising start has been made in the PATROLS project [220].

For SbD hazard testing, it is crucial to include tests that are predictive yet simple and cost-effective. Therefore, based on current literature, the use of a submerged co-culture model including at least a type of macrophage might be the most suitable. However, more research is needed to confirm that simple methods are predictive enough for early hazard screening.

### 4.5. Genotoxicity

One of the main safety concerns related to NMs is their possible genotoxicity [221,222]. Genotoxicity describes the capacity of a chemical or physical agent to produce genetic damage that, if left unrepaired, may lead to cancer [223]. Therefore, every mutagen is potentially carcinogenic [222].

Due to the important consequences to human health, mutagenicity is a hazard endpoint required in all product regulations (chemicals, biocides, pharmaceuticals, medical devices, food additives, cosmetics, etc.) [224]. The assessment of genotoxicity is based on validated in vitro assays, which can be followed up by validated in vivo assays, depending on the in vitro outcome and the regulation involved [225]. Therefore, genotoxicity assessment at an early stage of innovation is highly advised. In fact, genotoxicity is a key endpoint in most of the testing strategies developed for NMs [13,19,221,226,227].

#### 4.5.1. Most Frequently Used Assays, Strengths and Limitations

The mutagenicity of chemicals is usually evaluated on the basis of a battery of standard genotoxicity assays, able to detect gene mutations, chromosomal damage, and aneuploidy, as all these different mechanisms need to be considered in the assessment [221,224,227]. A core in vitro battery comprising the Ames test (detecting bacterial gene mutations) plus the in vitro micronucleus test (detecting chromosomal damage and aneuploidy) was already proposed 10 years ago for soluble chemicals [228]. Nearly 100% (958 out of 962) of rodent carcinogens or in vivo genotoxins were correctly detected with these two tests, which makes this battery a particularly sensitive combination [224]. However, the specificity of both assays together was unacceptably low (12.0%), giving rise to a high rate of false-positive results [229]. Hence, most of the EU regulations require a follow-up in vivo study when in vitro positive results are obtained [224]. In the case of NMs, the Ames test does not appear to be a suitable method as some NMs may not be able to penetrate through the bacterial wall, whereas others may kill the bacteria due to their bactericidal effects [230,231,232,233,234]. Based on this evidence, results obtained with this method should be followed up with other gene mutation assays using mammalian cells [21,235], or better yet, the Ames test should be avoided for NMs.

A roadmap for the genotoxicity testing of NMs was suggested some years ago [236], followed by guidance and common considerations [237]. There are two OECD TGs for assessing in vitro mammalian gene mutations: the In vitro Mammalian Cell Gene Mutation Tests using the Hprt and xprt genes (OECD TG 476), and the In vitro Mammalian Cell Gene Mutation Tests Using the Thymidine Kinase Gene (OECD TG 490). The latter, also called the mouse lymphoma assay (MLA), can detect a broader spectrum of genetic damage than the former, including chromosome rearrangements, deletions, and mitotic recombination [236]. Both assays are time-consuming, requiring long culture times (e.g., 10–14 days before counting colony formation), which has probably precluded an extensive use of these assays. For soluble chemicals, the sensitivity and specificity of the MLA assay was reported to be 73.1 and 39.0%, respectively, resulting in a prediction accuracy of 62.9 % [224]. In the case of NMs, given the low number of studies performed with these assays, and the wide variety of NMs included in these few studies, it has not been possible to draw any conclusions concerning the relative sensitivity of the various reporter genes to the potential mutagenicity of NMs [236]. Nevertheless, there are ongoing efforts to adapt the HPRT assay for use with NMs, e.g., within the EU H2020 RiskGone project, where round robin activities are ongoing.

Among the assays detecting chromosome damage, the In vitro Mammalian Cell Micronucleus (MN) Test (OECD TG 487) has been the most extensively used in nanotoxicology [221,230]. The assay detects chromosome mutations induced by either clastogenic or aneugenic agents. In addition, it also detects most mutagenic events as most mechanisms leading to gene mutation also induce chromosome mutations [228]. For soluble chemicals, the sensitivity and specificity of the MN assay was reported to be 78.7 and 30.8%, respectively [229], resulting in a concordance of 67.8%. In the case of NMs, there are few papers that evaluated a similar material, and there is a substantial variation in the methodology applied, which precludes raising conclusions on the reproducibility and predictability of this assay [236].

The most classically used version of the MN assay is the cytochalasin-blocked MN assay, which includes the use of cytochalasin B, a cytokinesis blocking agent that enables the identification of dividing cells [238]. However, as cytochalasin B may impair NM intracellular internalization, leading to false-negative results in the MN assay [239], it is recommended to successively treat the cells with NMs, and then with this agent [240,241]. Since cells should undergo mitosis for binucleated cells to form, the use of serum in exposure medium is recommended. The need of both proliferating cells and cells accumulating NMs in the MN assay has been illustrated recently while optimizing the MN assay on 3D cell models. The 3D EpiDerm™ skin model accumulates less NMs than 2D skin cells, resulting in less MNs upon exposure to genotoxic ZnO NMs [242]. Moreover, HepG2 spheroids still hold the capacity to proliferate while HepaRG spheroids do not, and genotoxic ZnO NMs do not show a positive outcome in the MN assay in 3D HepaRG while they do on the 3D HepG2 model [242,243]. An adaptation of this TG for NMs, within the OECD project 4.95 (‘Guidance Document on the Adaptation of In vitro Mammalian Cell Based Genotoxicity TGs for Testing of Manufactured Nanomaterials’), is currently ongoing based on previous recommendations [236]. One of the first round robin studies on the in vitro MN assay was performed within the NanoGenotox project [244], involving 12 laboratories, and comparing the genotoxicity of three reference materials. Relatively reproducible results were obtained in some cases, but they were material- and cell line-specific. A similar conclusion was reached by Louro et al. (2016) on four benchmark MWCNTs in two lung epithelial cell lines [245]. One reason could be the low fold increase over control values, as was also pointed out in the genotoxicity assessment performed by Elespuru et al. (2018) [236]. Currently, round robin tests are planned within the OECD project 4.95 and the EU H2020 RiskGone project.

Lately, the MN assay has been applied in co-culture systems involving inflammatory cells (e.g., THP-1 cells) and target cells (e.g., lung epithelial cells) allowing the evaluation of the mechanisms of action (primary vs. secondary) underlying NMs’ genotoxicity [246,247]. Genotoxins operating by a secondary mechanism of action, mediated by inflammation, are assumed to have a threshold response [248].

Classically, the MN assay has involved a labour-intensive manual scoring under the microscope. However, the speed of the analyses can nowadays be increased by using automated microscope scoring platforms [249], or flow cytometry [250,251,252]. The latter has recently been adapted to NMs [253].

The other validated method for assessing chromosome damage is the In vitro Mammalian Chromosome Aberration (CA) Test (OECD TG 473). This assay has been used much less because it is more time-consuming and requires a significant level of expertise to score the aberrations [236]. Furthermore, the CA assay does not detect aneugens, while the MN assay does (OECD TG 473, 2016). Hence, the CA assay would not be recommended for SbD hazard testing.

Furthermore, HTS approaches that could be applied to the testing of NMs are currently in development. These methods are non-OECD-guideline methods but proved efficient to detect potential NM genotoxicity. The comet assay is by far the most employed among these assays, and it could complement the recommended in vitro mutagenicity assays [236]. During the past decade, some effort has been dedicated to increase its throughput, with the highest achieved in the 96-minigel version using gel bond films [254]. It has been optimized and successfully applied to assess the genotoxicity of NMs within the FP7 NanoREG project [255].

Lastly, one commonly used genotoxicity assay for NMs is the immunolabelling of DNA repair protein foci, such as gamma-H2AX, which form during DNA double-strand break repair. The background of DNA double-strand breaks in cells is generally very low (although some exceptions exist, such as in some cancer cell lines), which makes these assays very sensitive. High-throughput versions of the assays exist. Foci can be counted using automated microscopy platforms or flow cytometry, with the advantage of their rapidity and possibility of analysing other cell parameters such as cell viability or apoptosis simultaneously [256]. Such high-throughput methods have rarely been applied on advanced 3D cell models for assessing NM genotoxicity because they necessitate additional steps. For example, a high-throughput comet assay can be performed on 3D cells after enzymatic and mechanical dissociation of the spheroid [257], reducing simplicity. Still, such advanced models could increase the predictivity of the assay. For example, Ag NMs cause significant DNA damage in a 2D liver-cell system, while the outcome of the comet assay is insignificant in 3D HepG2 spheroids, which is similar to most of the in vivo studies published up to now reporting Ag NM genotoxicity via comet assay [258].

#### 4.5.2. Overview of Needs and Knowledge Gaps

Table 5 shows a summary of how the different genotoxicity assays perform in terms of the criteria for SbD hazard testing. One of the main problems for determining the sensitivity and predictability of the genotoxicity assays when assessing NMs is the absence of nano-sized particulate controls. NM-specific controls have rarely been demonstrated [236,259], making comparisons among labs difficult. Furthermore, it is not possible to establish historical positive control ranges that would confirm the sensitivity of the tests [259]. Based on the currently available information, we follow the recommendations by Elespuru et al. (2018) [236] to use the MN assay in combination with a gene-mutation assay (HPRT or MLA). In the meantime, further optimizations of genotoxicity assays for testing NMs are ongoing.

For advanced models, such as 3D models, the 3D skin comet and micronucleus assays are sufficiently validated for conventional chemicals and individual OECD TGs could start being developed [260]. However, the 3D airway and liver models are still lacking assays that could measure micronuclei and gene mutations, respectively [260]. Working with NMs raises additional technical hurdles that need to be overcome. Nevertheless, advanced models will offer advantages over current assays, especially by mimicking better the human body response and being able to evaluate modes of actions, e.g., secondary genotoxicity [261].

Based on the currently available information, we follow the recommendations by Elespuru et al. (2018) to use the MN assay in combination with a gene mutation assay (HPRT or MLA). In the meantime, further optimizations of genotoxicity assays for testing NMs are ongoing.

## 5. Discussion and Outlook

The development, manufacturing, and use of NMs with novel properties and potentially undesired health risks are growing at a rapid rate. One way to reduce potential adverse effects caused by NMs is to incorporate SbD in NM development processes. Within a SbD approach, the potential hazards of a NM throughout the life cycle are assessed at an early stage of product innovation. Although advancements have been made in terms of nano-specific risk assessment strategies [13,19,138,227], several challenges remain when applying these strategies to SbD:Current hazard and risk-assessment strategies are not easily applied in an early hazard assessment for SbD applicability, as the proposed and required assays are too time-consuming and costly to be performed early in the development process of a NM.The suitability for SbD hazard testing of currently available in vitro assays in terms of predictivity, cost-effectiveness, sensitivity, specificity, robustness, and compatibility are largely unclear.

In this review, in vitro toxicity assays have been critically assessed in terms of their suitability for SbD hazard testing. The main purpose of SbD hazard testing is the identification of early hazard warnings and obtaining a general idea of the potential hazards of a novel NM, NEP, or components released thereof during the LC. It therefore serves as a first screening during the early stages of the development of a new NM or NEP. For SbD hazard testing, a balance needs to be sought between simplicity and comprehensive testing that addresses all concerns (Figure 5). The more elaborate the assessment, the more uncertainty is minimized, and the more the testing becomes too complex for the purpose of SbD.

### 5.1. Assay Predictivity

#### 5.1.1. Early Hazard Warnings

Correlating in vitro effects to in vivo potency has not yet been possible [109] and is not a requirement for SbD hazard testing. The identification of hazard warnings and detecting the most potent NMs is more relevant, and in this review it was shown that many assays are capable of doing this. The prediction accuracy of the evaluated assays in many cases depends on the type of particle, sample preparation, as well as the type of cell system used. Effects of NMs which assert their effect through ion shedding, such as Ag and ZnO NMs, were most accurately predicted across all toxicity endpoints. Current in vitro assays were less capable of predicting the effects of NMs acting through surface reactivity, and fibre-like NMs. Another important finding across several toxicity outcomes was the better predictivity of primary cells as compared to cell lines, as well as better predictivity of macrophage-like cells as compared to other cell types. However, there are clear indications that simple submerged assays might be suitable for prediction of adverse effects in vivo, as was also shown in a recent review by Di Ianni et al. (2022) [262].

#### 5.1.2. Hazard Ranking

For an assay to be able to establish an accurate ranking in toxicity is valuable for SbD hazard testing, as this allows the use of the assay for comparison between candidate NMs, and for comparison to benchmark materials with known toxicity. There are indications that in vitro cytotoxicity assays are able to predict an adequate ranking in toxicity which corresponds to in vivo pulmonary inflammation [114]. Also, the detection of cytokines at the ALI could potentially detect a ranking that corresponds to in vivo pro-inflammatory mediators [211]. However, both of these studies could not draw any definitive conclusions on comparable rankings. Two studies showed that simple submerged cell lines are able to produce accurate rankings in pro-inflammatory effects that corresponded to in vivo markers of fibrosis [208,209]. It is important to note that the accuracy of toxicity rankings is hugely dependent on the calculation of the dose delivered to the cells, and this should therefore always be carried out [59].

#### 5.1.3. Applicability Domains

A low prediction accuracy of an assay could be improved by exploring the exact applicability of the assay. Ensuring that the specific MOA that caused the in vivo toxicity can be detected using the assay will reduce the rate of false negative outcomes. The applicability domain of each assay should be well-understood (which assay can predict what kind of in vivo (human) toxicity) to be able to use assays that are fit-for-purpose. When looking at the transition to animal free testing in general, this is one of the issues that needs addressing for soluble chemicals as well [263]. In that respect, in vitro toxicity testing of NMs should be mechanism-based by looking for specific effects or MOAs [264]. If the applicability domain of assays and cell models is established with more certainty, it is possible to combine assays into a strategy to holistically assess NM toxicity in vitro. In such a strategy, assay and cell type selection would be facilitated by a combination of applicability domains and the most relevant exposure route.

#### 5.1.4. Prediction Accuracy

Prediction accuracy in terms of sensitivity and specificity should be determined for more assays, to increase knowledge about assay reliability. Determination of accuracy is a crucial step during the validation process of in vitro assays in general [265]. For example, for regulatory genotoxicity testing, it is known that in vitro assays have a high sensitivity but low specificity. This means that there is a high chance of false positives, and positives should always be confirmed in an in vivo study. A better understanding of the prediction accuracies of in vitro assays used for SbD hazard testing would help enormously with the interpretation of results.

#### 5.1.5. Challenges in Assessing Predictivity

Although in vitro assays have been used for some time to test NM toxicity, not all the criteria could be evaluated properly due to the lack of or limited availability of high-quality data. Especially for predictivity, there is a data gap that needs to be filled in order to correctly interpret results. Several factors complicate the assessment of prediction accuracy of in vitro assays. Since there is a lack of human data on NM toxicity, predictivity of assays is at present evaluated compared to in vivo data derived from studies in experimental animals. This means that an in vitro model comprised of human cells is being compared to animal data, to predict a human response. The relevance of this approach is questionable, due to the differences between humans and experimental animals [266,267]. The lack of deposited dose calculations, interference controls, proper characterization, and varying sample preparation protocols (e.g., the use of serum, different media, different dispersion techniques) across the literature add another layer of complexity to assessing the predictivity of assays. Since these factors can have such an impact on assay outcome, assay standardization will aid in determining assay predictivity for adverse human health effects. Additionally, the lack of clear positive and negative controls for NMs hampers the assessment of prediction accuracy.

Finally, it must be noted that in the papers reviewed here, an optimistic perspective is given about the predictivity of in vitro assays for toxicity and adverse health outcomes in vivo. We should however be cautious, as negative results or results with a low correlation with known in vivo or human health responses may not reach publication: a phenomenon known as publication bias.

### 5.2. Outlook for Innovators, Regulators, and Industry Based on Current Knowledge

An overview of the most important knowns and unknowns with regards to NM SbD hazard testing is summarized in Table 6. Figure 6 shows what we think is the road forward towards successfully putting SbD hazard testing into practice. The successful implementation of SbD hazard testing requires efforts from innovators, regulators, as well as from industry.

#### 5.2.1. A Change in Mindset towards Purpose-Driven Innovations

The current European policy landscape (the European Green Deal, the European Chemical Strategy for Sustainability and the Zero Pollution Action Plan [2,3,268]) demands a new mindset for innovating. SbD provides an approach aiming at developing safer NMs and NEPs by integrating safety into the innovation process and material development in a LC thinking approach, from design to end-of-life. Any innovation that does not have a green or sustainable purpose will not survive.

#### 5.2.2. Starting In Silico: Databases and SARs

SbD hazard assessment should first and foremost be based on material knowledge and material–activity relationships. Before even starting in vitro experiments, an elaborate evaluation of available physicochemical data should be performed [220]. Here, certain hazard warnings could already be noticed. For example, the structure–activity relationship (SAR) of high aspect ratio NMs (HARNs) and mesothelioma risk is widely accepted [138]. It would be unnecessary to perform hazard testing on HARNs, as it would already be clear beforehand that this material raises a hazard warning. Another potential hazard warning would be respirable crystalline silica particles, due to their structure–activity relationship with silicosis and lung cancer [269,270]. Knowledge on structure activity relationships is especially important for the identification of potential hazards and application of SbD interventions.

For novel advanced materials however, limited information on these tox-driving properties is available, and the SbD decisions are mostly based on SbD hazard-testing outcomes.

#### 5.2.3. Importance of Experimental Design

The physical aspects of NMs add another dimension to the complexity of toxicity testing. It should always be considered that the way the experiment is carried out (dispersion protocol, medium type, addition of serum) affects the outcomes and that the behaviour of the particle in the culture dish (settling, agglomerating, floating, dissolution, formation of protein corona) should always be analysed [23,49,53,58,59]. Checking and accounting for assay interference is crucial, also for SbD hazard testing and high throughput screening, where it is often overlooked [97]. This makes SbD hazard testing for NMs more challenging than that of soluble chemicals.

#### 5.2.4. Combinations of Assays

An integrated approach to testing and assessment (IATA) that can combine information from multiple sources (available data, in silico tools, in vitro assays) is the way forward towards an effective early hazard identification of NMs and NEPs and for the development of SbD interventions. This review discusses simple assays since it focusses on the initial stages of innovation. However, at more advanced stages, SbD hazard testing may also include approaches that are not as simple and cost-effective [264]. With regards to the transition to animal-free alternatives, the focus on simplicity as is required for SbD hazard testing should not create a barrier for the development of more realistic and innovative cell models with potentially better predictivity, such as induced pluripotent stem cells (iPSCs) and organoids.

For inflammatory potential, chronic inflammation (leading to tissue damage and remodelling as well as loss of functions) is the adverse outcome of concern, which is presently not captured by one or more in vitro tests. An acute pro-inflammatory effect in an in vitro assay as measured by cytokine secretion, in combination with slow dissolution, indicating high bio-persistency [14], might together indicate that the NM induces chronic inflammation. Combining assay outcomes in SbD hazard testing should be further explored.

#### 5.2.5. Thresholds for Toxicity

In order to raise hazard warnings and to interpret results from combinations of assays, thresholds are needed. This is especially challenging for inflammatory potential assays. Macrophages are the major defence mechanism against foreign materials, and their activation is crucial for the clearance of NMs [192]. It is unclear when a beneficial immune response turns into persistent pulmonary inflammation in vivo, and how to predict this in vitro.

Previously established frameworks have made a step towards generating thresholds for toxicity. The Nanoreg2 framework and the Swiss Precautionary Matrix score NMs as low, medium, or high hazard according to their fold change increase as compared to a negative control [271]. The Nanoreg2 framework adds a scoring system that allows for combining outcomes of different assays, and subsequent comparison of different NMs. With both approaches, a significantly positive response in an assay might still lead to a classification as low hazard.

Since SbD hazard testing is performed as an early screening, and its main goal is determining early hazard warnings, a zero-tolerance principle might be more suitable in this case (as is common practice in the pharmaceutical industry). For primary genotoxicity, a zero-tolerance principle is already in place in regulatory risk assessment, as genotoxic carcinogens are regarded as having no threshold and thus an acceptable exposure level cannot be derived [224]. For SbD hazard testing, it could be argued that a worst-case approach would be suitable for the other endpoints as well, meaning that any indication of inflammation, reactivity, or cytotoxicity at relevant doses would raise a hazard warning. Here it is important to consider the possibility of false negatives produced in the assays.

For SbD hazard testing, the inclusion of benchmark NMs with known in vivo toxicity is recommended to compare the new NM to existing information. Thresholds could be set according to the response of the benchmark NM in a specific assay. Alternatively, an appropriate ranking in potency of NMs could be useful for making SbD decisions when comparing several candidate NMs.

#### 5.2.6. Assay Standardization

In Figure 6, assay standardization is represented connecting many important aspects. As mentioned throughout this review, assay standardization is a key need for the further development of SbD hazard testing, as well as for putting SbD into practice. Firstly, we showed that in vitro-in vivo comparisons are hampered by the lack of standardized protocols. Moreover, fundamental research into structure–activity relationships will benefit from standardized protocols as well. Assay standardization will result in more high-quality fundamental data on the MOAs of toxicity of NMs, which will in turn aid the refining of SbD hazard testing. Ultimately, standardization will increase the chances of industrial use and acceptance of these assays into existing legal frameworks, which will make incorporating SbD approaches more appealing for manufacturers [272].

On the contrary, the complexity of NM toxicity testing hampers the standardization of assays. It is for example impossible to create one exposure method suitable for all NMs, especially considering NMs of the future which will possess yet unknown properties. A case-by-case or targeted approach will be needed for specific NMs with incompatible PC properties. In some cases, standardization may not be feasible, but guidance will be of great help.

Assay standardization should be followed by assay validation in order to improve our understanding of the robustness, predictivity, and compatibility of the assays. The ongoing work in the OECD’s Working Party on Manufactured Nanomaterials [273], the Malta Initiative [274], and work in ongoing European projects such as NanoHarmony [275], Nanomet [276] and Gov4Nano [277] are currently supporting the standardization efforts.

#### 5.2.7. Compatibility (NEPs and Novel Materials)

Safety along the LC as well as keeping pace with the rapid emergence of advanced materials are important hallmarks of SbD. Consumers are most likely exposed to NEPs and not pristine NMs. Therefore, assay optimization is needed to be able to test NEPs in an accurate way. Assay compatibility with NEPs and novel advanced materials needs to be studied further. More data on how NMs can change over the LC and the possible risks they may pose during this process is very much needed. This will help determine whether testing only pristine NMs may be sufficient for SbD hazard testing.

#### 5.2.8. Gathering Experimental Data following FAIR Principles

Since SbD hazard testing will involve the generation of large datasets, it is important to ensure that the data gathered from the different in vitro assays are adequately collected using templates that support FAIR principles, and that the data is findable, accessible, interoperable and reusable. Guidance for finding these templates can be found in the GoFair initiative and guidance on experimental workflows design and implementation can be found within the NanoCommons initiative.

#### 5.2.9. The Chemical Strategy for Sustainability

Although this review covers cytotoxicity, dissolution, oxidative potential, inflammatory potential, and genotoxicity, the Chemical Strategy for Sustainability has put forth extra endpoints to ensure the ambition towards a toxic-free environment and protection against the most harmful chemicals is fulfilled [3]. One of these endpoints is endocrine disruption. Under REACH, endocrine disruptors are identified as substances of very high concern alongside chemicals known to cause cancer, mutations, and toxicity to reproduction. Work is ongoing by ECHA to develop classification and labelling criteria for endocrine disruption [278]. From a NM-perspective, there is increasing evidence showing endocrine disruption and reproductive impairments caused by NMs such as nano plastics [279,280], and this warrants further attention.

Although this review is only focused on SbD, sustainability impacts should also be considered early in the innovation process. Safe-and-sustainable-by-design is a central element of the European Chemical Strategy for Sustainability and it demands the optimization of safety and sustainability interventions in the design of NMs, NEPs, and all processes in a life-cycle approach.

## 6. Conclusions

This review provides the first building blocks towards an early hazard testing strategy for SbD applicability and is the first detailed state of the art analysis of in vitro assays against performance criteria (simplicity and cost effectiveness, predictivity, robustness, compatibility, and readiness) for SbD hazard testing. The most important conclusions are:Based on current knowledge, primary cell models and more physiologically relevant exposure methods provide better predictions of in vivo results. However, the aim of SbD hazard testing is to detect early hazard warnings using simple methods. There are strong indications that simpler assays, such as acellular OP assays, static dissolution assays, and simple submerged cell-based assays for cytotoxicity, genotoxicity, and inflammation give sufficiently accurate information for identifying early hazard warnings or even hazard rankings, when carried out correctly.The suitability of these simple assays for SbD hazard testing has to be further confirmed in future studies. More model comparisons between simple, complex, and in vivo models are needed to investigate whether simple in vitro models are indeed sufficiently predictive and suitable for SbD hazard testing, preferably using standardized methods. Additionally, the applicability domain of in vitro assays to detect NM toxicity should be mapped more precisely to correctly interpret results.Assay standardization proved to be critical for the progression of SbD hazard testing as it will improve in vitro-in vivo comparisons, improve fundamental knowledge on NM toxicity, support industrial use, and is a first step towards regulatory acceptance.Simplicity is not always feasible when testing NMs, even though it has been put forward as one of the criteria for SbD hazard testing. Dispersion protocols, dose delivered to cells, compatibility issues, interferences, testing NEPs and NMs released along the LC, etc., all complicate SbD hazard testing of NMs and reduce achievable simplicity. Innovators, industry, regulators, and policymakers should realize that the hazard assessment of NMs and advanced materials is complex and that in vitro tests need to be further developed, tested, and evaluated to assess their suitability in identifying potential hazards.

## Figures and Tables

**Figure 1 nanomaterials-13-00472-f001:**
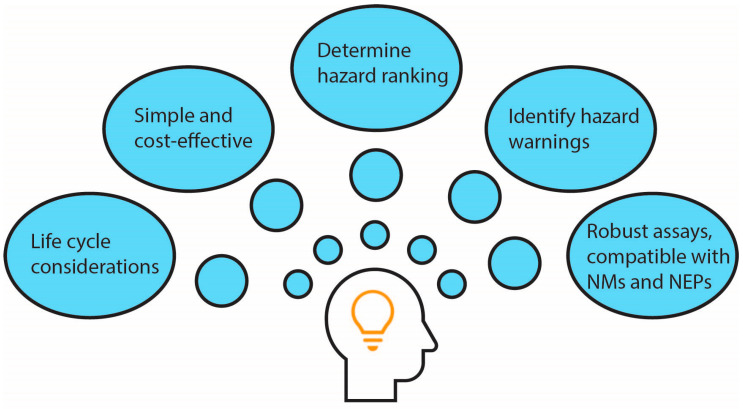
Considerations for assay selection for SbD hazard testing.

**Figure 2 nanomaterials-13-00472-f002:**
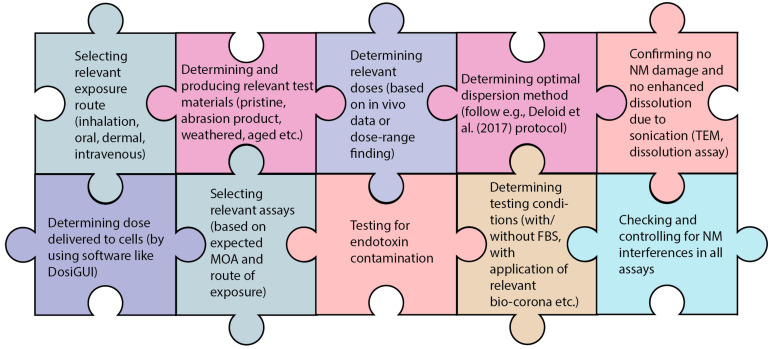
Overview of aspects that might have to be considered when performing SbD hazard testing, showing that simple testing can be challenging to achieve.

**Figure 3 nanomaterials-13-00472-f003:**
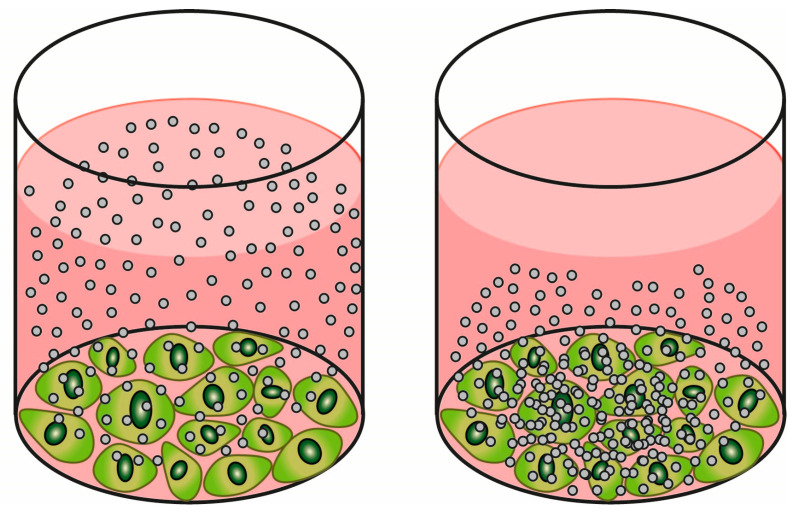
Visual representation of two NMs with different properties, resulting in different doses delivered to the cells, when administered doses are equal.

**Figure 4 nanomaterials-13-00472-f004:**
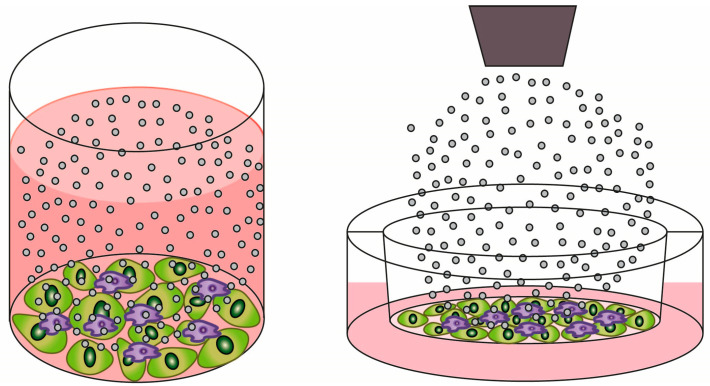
Submerged (**left**) and ALI exposures (**right**) to NMs. Submerged exposures are considered easier, whereas ALI exposures are considered more physiologically relevant for inhalation (and dermal and intestinal) exposures.

**Figure 5 nanomaterials-13-00472-f005:**
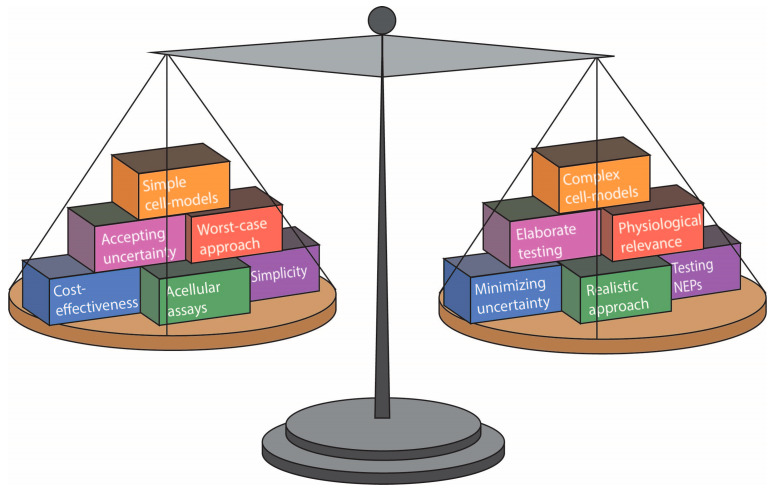
The balance of SbD hazard testing. SbD aims to address safety at an early stage in the product development process. On the one hand, SbD tries to be comprehensive to address all concerns, while on the other hand the approach should be simple.

**Figure 6 nanomaterials-13-00472-f006:**
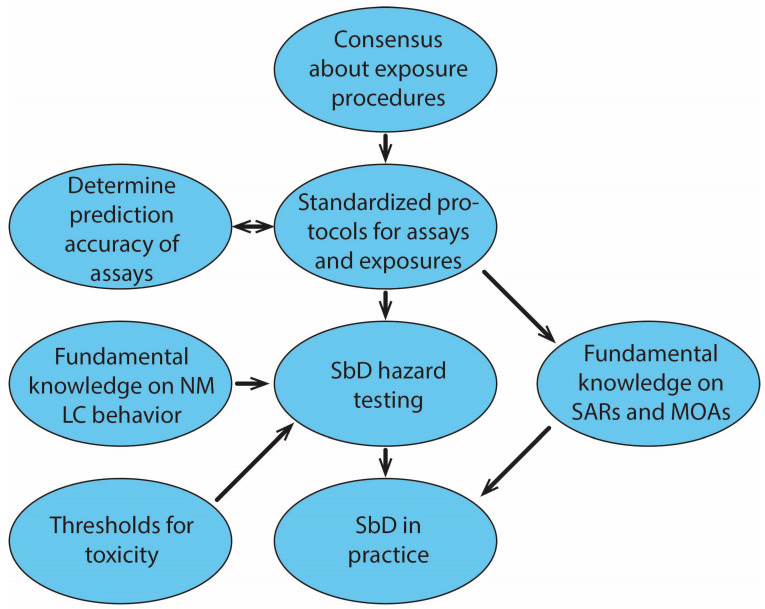
Factors that became evident throughout this review that are crucial for putting SbD hazard testing into practice. Protocol standardization is key for SbD hazard testing, as well as for better understanding structure–activity relationships and prediction accuracies of assays.

**Table 1 nanomaterials-13-00472-t001:** Evaluation of suitability of cytotoxicity assays for SbD hazard testing.

Performance Criteria	Mitochondrial Activity (MTT, MTS, XTT, WST-1, Alamar Blue)	Cell Membrane Integrity (LDH)	Cell Membrane Integrity Staining (Trypan Blue, Propidium Iodide, Annexin V)	Lysosomal Integrity (Neutral Red Uptake)	Caspase 3/7 Assay
**Simplicity and cost**	 Easy and cost-effective, commercial kits available.	 Easy and cost-effective, commercial kits available.	 Microscopic evaluation is time-consuming. Using flow cytometry increases time efficiency.	 Easy and cost-effective, commercial kits available.	 Easy and cost-effective, commercial kits available.
**Predictivity (Sensitivity and Specificity)**	  Depends on the mechanism of toxicity of the particle, and the cell type used [110]. Macrophages seem more sensitive [114,117]. Assay better equipped to detect cytotoxicity of ion-shedding NMs [100]. Possibly suitable for making accurate rankings in toxicity [114].	  Depends on the mechanism of toxicity of the particle, and the cell type used. LDH results have been shown to correlate with in vivo results for ion shedding NMs [112] as well as poorly soluble NMs [113].	 Not assessed for NMs specifically.	 Not assessed for NMs specifically	 Not assessed for NMs specifically.
**Robustness**	 For MTS assay, decent robustness but depending on cell type used [107], and only when interferences are correctly avoided [100,108]. More elaborate SOPs and harmonization between labs enhance assay robustness [52,102].	 Similar robustness as MTS assay [108].	 Not assessed for NMs specifically.	 Not assessed for NMs specifically.	 One study showed high inter-laboratory variability [100].
**Compatibility**	 Many NMs interfere with the substrate, the product, or the optical readout. Can be overcome by washing cells before incubation with reagent, and centrifugation to get rid of NMs [100,108].	  Many NMs interfere with the enzyme, the reagent, or the optical readout. Can be overcome via centrifugation. Washing not possible as LDH is measured in supernatant.	  NMs may interfere with the dye.	  NMs may interfere with the dye.	  NMs may interfere with the dye or the readout.
**Readiness**	 ISO protocol for MTS assay.	 No NM-specific standardized protocol available.	 No NM-specific standardized protocol available.	 No NM-specific standardized protocol available.	 No NM-specific standardized protocol available.

**Table 2 nanomaterials-13-00472-t002:** Evaluation of suitability of dissolution assays for SbD hazard testing.

	Acellular Assays	Cellular Assays
Performance Criteria	Static Dissolution (e.g., OECD Series on Testing and Assessment No. 29)	Flow-through/Dynamic Dissolution	Cellular In Vitro Dissolution
**Simplicity and cost**	  This system is the simplest and could be conducted by commercial laboratories without extensive investment in equipment.	 Requires much greater effort with regards to setup, and also requires a large volume of fluid.	 The basic principle of this method is simple and can be performed cheaply. Would be considered high throughput but as cellular will typically incur higher costs than acellular.
**Predictivity (Sensitivity and Specificity)**	  Static dissolution studies have been found to correlate with in vivo results in some instances [143,146,151], but in others poor correlation is observed [137,144,146,152,153]. Losses in sensitivity may arise due to any sample handling (e.g., acidifying the sample, filtration) or saturation of ions. For highly soluble materials, dissolution may continue during centrifugation steps, resulting in greater values of dissolution.	  Good correlation observed between flow-through system using a specific simulant fluid (modified Gamble’s) and intratracheal instillation in vivo [154], and for some particles dynamic dissolution in phagolysosomal simulate fluid (PSF) was a good predictor for short term inhalation study in rats [123] and intratracheal instillation in rats [137]. Losses in sensitivity may arise due to any sample handling (e.g., acidifying the sample). Additional concerns about losses in the system due to filtration.	  Results do appear to correlate well with in vivo in some instances (e.g., fast dissolution of Ag NMs in vivo [155] and in vitro [149]). Study by Koltermann-Jülly et al. (2018) found very low levels of dissolution in macrophages compared with the abiotic flow-through system and clearance in vivo [123]. Sensitivity relies on the capability of analysing released material. Additional concerns may arise from complexing of ions to biomolecules.
**Compatibility**	  The basic setup is compatible with many materials. Issues may arise with hydrophobic materials and with any material whereby sensitivity cannot be achieved for further analysis due to interference with components in the biofluid mixture (e.g., Ag NMs).	  The basic setup is compatible with many materials. Issues may arise with any material whereby sensitivity cannot be achieved for further analysis due to interference with components in the biofluid mixture or membranes used (e.g., Ag NMs).	  Most common analytical technique used is ICP-MS, therefore this methodology is the most compatible with metals. Carbon-based NMs such as CNT have used analytical techniques such as UV-Vis, Raman spectroscopy, and EM, however the sensitivity of these techniques is likely to be far less.
**Robustness**	 Large variability between different biofluids.	  Can result in false positives and false negatives due to issues with the filtering system (i.e., due to NMs passing through pores or causing blockages in filters).	 No evidence of inter-laboratory comparisons. Issues may arise due to inclusion of particles on the surface of the cell rather than internalised particles only.
**Readiness**	  OECD protocol but specifically for environmental studies. Various fluid compositions available.	  ISO protocol outlining basic methodology. TRL identified as high/medium for metals and medium/low for organic materials (e.g., CNT) [124].	 Validated assays available but no standardized method.

**Table 3 nanomaterials-13-00472-t003:** Evaluation of suitability of oxidative potential assays for SbD hazard testing.

Performance Criteria	FRAS	ESR/EPR	DCFH Acellular	Haemolysis Assay
**Simplicity and cost**	  Very simple but needs large amounts of NM.	  Very simple, yet might be difficult to find lab with specialized ESR/EPR equipment.	  Very simple and only requires a fluorescence reader.	  Very simple and only requires absorbance reader and whole blood.
**Predictivity (Sensitivity and Specificity)**	 The assay is able to detect NMs’ reactivity at low concentrations and in a dose-dependent manner with higher sensitivity compared to DCFH assay [163]. Could distinguish between CNT types [175]. Prediction accuracy reported: 50% [162].	 Depending on spin trap used. Aids to identify specific ROS types, which could be useful for SbD interventions [176]. Prediction accuracies reported: 69% [164] and 50% [162]. Correlated well with in vitro cytotoxicity and protein carbonylation [183,184].	  Lacks sensitivity as compared to FRAS and ESR/EPR [163,171,175]. However, protocol adaptations [172] show ameliorated sensitivity. Prediction accuracies reported: 77% [164].	 Is thought to be able to detect OP of both surface reactive as well as ion-shedding NMs [112]. Showed very high prediction accuracy (92%) in one study [164].
**Compatibility**	  Good compatibility with a wide range of NMs. Optical interferences are largely avoided using a centrifugation step but have been reported [173]. Adapted method suggested for graphene-based materials [168].	  Good compatibility with a wide range of NMs [162]. No interferences reported.	 High background signals resulted from dye auto-oxidation [171]. NM interferences reported [162,166,184]. Adapted DCFH protocol reduces interferences [172].	 No interferences reported, yet might be expected due to absorbance readout.
**Robustness**	 No interlaboratory study performed. Found to be reproducible and reliable within the same lab [163,165]	 Not assessed for NMs specifically.	  Previously lacked robustness [175]. Interlaboratory round robin tests in GRACIOUS project showed satisfactory reproducibility for positive control NMs using optimized SOP [170].	 Not assessed for NMs specifically.
**Readiness**	  No NM-specific standardized protocol available. Gandon et al. (2017) protocol available [163].	 ISO protocol available (ISO 18827:2017)	  No NM-specific standardized protocol available. Boyles et al. (2022) protocol available [170].	 No NM-specific standardized protocol available.

**Table 4 nanomaterials-13-00472-t004:** Evaluation of suitability of Inflammation assays for SbD hazard testing.

	Submerged Cell Models	ALI Cell Models
Performance Criteria	Submerged Cytokine Release Mono-Culture	Submerged Cytokine Release Co-Culture	ALI Cytokine Release Mono-Cultures	ALI Cytokine Release Co-Cultures
**Simplicity and cost**	 Simple and cost effective.	 Simple and cost effective, however creating a co-culture requires more effort and experience than a mono-culture.	 Requires specialized exposure equipment and a certain level of expertise.	 Requires specialized exposure equipment and a certain level of expertise; creating a co-culture requires more effort and experience than a mono-culture.
**Predictivity (Sensitivity and Specificity)**	 Good correlation with in vivo found [190]. Found to be more sensitive than ALI in several studies [214,216]. Accurate ranking found [208,209].	 Combination of immune cell and epithelial cell more predictive than epithelial cell alone [94]. Accurate ranking found [213].	 Generally good for primary cells. Lower predictivity of epithelial cell lines. ALI exposures found more sensitive than submerged in several studies [210,211,212].	 Co-cultures perform better than two cell types separately [94]. BMDL of this model comes closer to the in vivo BMDL compared to submerged [211].
**Robustness**	 Large inter-laboratory variability for THP-1 cells [100,108], no inter-laboratory data on other cell types.	 Not assessed for NMs specifically.	 Low reproducibility but similar trends between labs [195]. Low reproducibility improved after protocol optimizations [196].	 Low reproducibility but similar trends between labs [195].
**Compatibility**	 NMs may interfere with ELISA [99,118].	 NMs may interfere with ELISA [99,118].	  Compatible with a wide range of materials, including hydrophobic and low-density NMs. However, NMs may interfere with ELISA [99,118].	  Compatible with a wide range of materials, as exposures do not necessarily require a dispersion. However, NMs may interfere with ELISA. Might be more suitable for NMs released form NEPs.
**Readiness**	 No NM-specific standardized protocol available.	 No NM-specific standardized protocol available.	 No NM-specific standardized protocol available.	 No NM-specific standardized protocol available.

**Table 5 nanomaterials-13-00472-t005:** Evaluation of suitability of genotoxicity assays for SbD hazard testing.

	Simple Cell Models	More Complex Cell Models
Performance Criteria	Gene Mutations in Cell Lines (OECD TGs 476 and 490)	Chromosome Damage in Cell Lines (OECD TGs 487 MN Assay)	Gene Mutations in Advanced Models	Chromosome Damage in Advanced Models (OECD TG 487)
**Simplicity and cost**	  Time consuming, requiring long culture times (e.g., 10–14 days before counting colony formation). Relatively cheap.	 Simple and relatively cheap. Analyses can be sped up using automatic image analysis systems and flow-cytometry.	 Not used up to now, would necessitate 3D model dissociation before cell plating, i.e., simplicity reduced as compared to simple models.	  Relatively simple and cheap for advanced models. Would be more time consuming and expensive than 2D models [260].
**Predictivity (Sensitivity and Specificity)**	 Conventional chemicals: adequate (62.9%) [224,229]. NMs: no conclusions can be reached [236,259].	 Conventional chemicals: adequate (67.8%) [224,229] NMs: no conclusions can be reached [236,259].	 Not used up to now, no conclusion can be reached.	 Co-culture systems may allow the evaluation of the involved genotoxicity mechanisms of action [246,247]. They may be more predictive of an in vivo-like response [260]. 3D models do not seem to be appropriate for applying this assay due to the lack of cell proliferation [242,243]. When the 3D model involves proliferating cells, it is more sensitive than 2D models, due to higher metabolic activity [260].
**Robustness**	 No inter-laboratory comparisons available for NMs. Ongoing comparisons within the EU H2020 RiskGone project.	  Relatively reproducible results in some cases, but material- and cell line-specific [244]. Future inter-laboratory comparisons under the OECD project 4.95.	 Not used up to now, no conclusion can be reached.	 Not enough studies available yet to allow reaching conclusions.
**Compatibility**	 Too low number of studies to reach conclusions [236].	 Suitable for different NMs (no interferences reported). No information about adequacy for complex materials.	 No conclusion can be reached. Still, for NMs could prove unsuitable since only the cells at the periphery of the spheroid/organoid would be exposed to NMs.	 Suitable for different NMs (no interferences reported). No information about adequacy for complex materials.
**Readiness**	 No NM-specific standardized protocol available.	 No NM-specific standardized protocol available.	 No NM-specific standardized protocol available.	 No NM-specific standardized protocol available.

**Table 6 nanomaterials-13-00472-t006:** Overview of the most important findings in this review, including knowns and needs for SbD hazard testing.

		What We Know for SbD Hazard Testing	What We Need for SbD Hazard Testing
**NM treatment**	**Dispersion protocols**	-Sonication can destroy intrinsic NM properties that might be part of its safer design. -Sonication can induce underestimation of toxicity by reducing the length of CNTs. -Sonication can enhance dissolution and release of (toxic) ions. -Sonication leads to a lower state of agglomeration.	-Consensus around dispersion protocols. -Dispersion guidance which covers all relevant exposure conditions and takes into account SbD interventions.
	**Experimental design**	-Testing NMs with serum results in lower in vitro toxicity. -Calculation of the dose delivered to the cells can have impact on toxicity ranking and is therefore also required for SbD hazard testing.	-Consensus and guidance for experimental design in the context of SbD.
	**Compatibility and LC**	-Humans are not only exposed to pristine NMs, but also to NEPs, aged NMs, and NMs released during the LC. -Testing NMs released from NEPs may pose challenges in terms of feasibility and compatibility -Compatibility of novel NMs with currently available in vitro assays unknown.	-Guidance on how to approach testing NMs with unknown compatibility. -More research towards determining whether testing pristine NMs is sufficient for SbD hazard testing.
**Assay protocols**	**Cytotoxicity**	-More elaborate SOPs enhance robustness. -Many NMs interfere with cytotoxicity assays, which should not be overlooked. -In vivo effect of ion shedding NMs is sufficiently accurately predicted. -Measuring cytotoxicity is useful for identifying hazard warnings.	-Further standardization and validation of cytotoxicity assays -Thresholds for cytotoxicity in the context of SbD -More focus on assays that do not pose interference issues. -To confirm predictivity of cytotoxicity assays
	**Dissolution**	-Dissolution rate may infer bio-persistency, which is important information for SbD hazard testing. -Predictivity largely depends on readout method as well as biological fluid choice. -Static acellular dissolution seems to be the most appropriate method for SbD hazard testing, especially as they are rather simple, however some studies indicate otherwise.	-To confirm that measuring static acellular dissolution is indeed sufficiently predictive for SbD hazard testing. -Meaningful thresholds for dissolution rates that allow for detection of differences that will lead to meaningful SbD decisions and interventions.
	**Oxidative Potential**	-Acellular assays might be predictive enough for SbD hazard testing. -In vivo effects of ion-shedding NMs is sufficiently accurately predicted. -There are indications that the haemolysis assay can accurately predict effects of surface-reactive NMs. -FRAS and ESR assays are more sensitive than DCFH. -FRAS assay can provide accurate ranking.	-To confirm that measuring acellular OP is predictive enough for SbD testing. -Meaningful thresholds for OP in the context of SbD.
	**Inflammation**	-The use of a type of immune cell is crucial (using only epithelial cells is not sufficient). -Primary cell models have better predictivity, but (immune) cell lines may suffice for SbD hazard testing. -Co-cultures seem to perform better than mono-cultures. -More elaborate SOPs enhance robustness.	-More work needed to develop in vitro models that can predict chronic inflammation. -Thresholds for in vitro inflammation in the context of SbD. -To confirm that submerged mono-cultures of macrophage cell lines are predictive enough.
	**Genotoxicity**	-Prediction accuracies very well established for soluble chemicals, but not for NMs. -It is important that the cell model of choice is capable of NM uptake. -The absence of NM positive controls makes determination of prediction accuracy challenging.	-To determine prediction accuracies of assays for NM specifically. -Round robin initiatives to test robustness of assays.

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
