# Peer review of "The State of the Art and Challenges of In Vitro Methods for Human Hazard Assessment of Nanomaterials in the Context of Safe-by-Design"

_nanomaterials, 2023, doi:10.3390/nano13030472_

Round 1

Reviewer 1 Report

The review "The state-of-the-art and challenges of in vitro methods for human hazard assessment of nanomaterials in the context of safe-by-design" concerns the safe-by design (SbD)concept and various in vitro methods used during hazard testing of new materials/products. Since the SbD concept is applied mainly in nanotoxicology the review also focused to potential problems in nanomaterial assessment. In vitro methods reviewed in the manuscript are as follows: cytotoxicity, dissolution oxidative potential and stress, inflammation, genotoxicity.

The review is very comprehensive and although the manuscript is well organized I found it difficult to read. The aim of the Authors was to describe so many toxicological endpoints that some topics, in my opinion, have been covered very superficially, and some of them are repetition of existing knowledge. Among the references I was unable to find some existing reviews, e.g. the review on assessing the genotoxicity of nanomaterials (Dusinska et al., 2019 / Methods Mol Biol. 2019;1894:83-122).

The Chapter 3.2. -  maybe it is worth mentioning the tendency to replace animal products with animal-free products?
Is it essential to prove that NMs are well dispersed during in vitro exposure? How can sedimentation of poorly dispersed NMs affect dose or effect determination?

I would suggest narrowing the scope of the manuscript and exploring topics that have so far been poorly covered.

Reviewer 2 Report

This long review manuscript titled “The state-of-the-art and challenges of in vitro methods for human hazard assessment of nanomaterials in the context of safe-by-design” by Nienke Ruijter et al. provides the building blocks that towards an early hazard testing strategy for Safe-by-Design (SbD) applicability. They state the analytical methods for in vitro assays of SbD against performance criteria ones which is very interesting and might be valuable and applicable for the future study. However, the basis of effective nanomaterial risk assessment is very complex though the research has been investigated for nearly two decades, the criteria of analytical processes and methods, and the reporting standardization that can be achieved are still ongoing discussed.

Some comments and suggestions:

1. In section 3, the structure of the text seems not very clear. It discussed the in vitro processes and assay methods, thus the methods should dependents on the exposure rout and the relevant target organs, thus the co-culture or ALI exposure can be chosen, further, the typical cell types should be selected. In this case, the stability of the nanoparticles such as aggregation, composition, crystallinity, shape, size, and surface chemistry changes should be tested. The subtitle “The use of serum and stabilizers” seems inappropriate that does not exactly reflect the possible changes and specific exposure route.

The hazard analysis of nanomaterials in their life cycle is important, but the release property is only one issue, the changes of oxidative states on nanosurface might be even more important, similarly with the asbestos, silica.

2. The big difference of hazard testing between engineered nanomaterials and chemical compounds is that the nanomaterials compose various structures including size, shape (including two-dimensional shape, etc.), chemical composition, crystal structure, surface chemistry (charge, chemical ligands, defect, etc.), etc., thus the toxic test is more complex and the analytical methods will sometimes infeasible when use the criteria ones. However, the article only mentioned the dissolution of nanomaterials. In addition, the oxidative potential induction is highly associated with the physicochemical properties of nanomaterials and the various environments they exposed, thus the alternatively analytical methods should dependent on these situations.

3. I suggest the authors rearrange the structure of the text. Some topics appeared repeatedly in several times, such as dissolution of nanomaterials.   

    Overall, I hope the review article could discuss the alternative or optimal analytical methods dependents on the physicochemical properties, the exposure pathway and the targeted organs or cells, etc. instead of general discussion of the methods.

Reviewer 3 Report

The manuscript is well written with important scientific information and well organized with relevant data. However, I would say it is a bit long and some sections could be summarized but it's up to you.

Round 2

Reviewer 2 Report

The Horizon2020 project SAbyNA that aims to develop a user-friendly platform for industry with optimal workflows to support the development of Safe-by-Design (SbD) of nanomaterials and nano-enabled products (NEPs). This work is done under the project that try to provide a practical and critical evaluation of the suitability of most frequently used in vitro toxicity assays for NM SbD hazard testing. For this purpose, the work is interesting and very important. As it is known the basis of effective NM risk assessment is very complex that the standardizations which can be achieved are still ongoing discussed, this review article also made some discussion about the challenge.

I appreciate the revisions of the manuscript. The authors have made great improvement of the article for more clearly presenting the methods for appropriate use in the in vitro toxic testing of NM SbD. I suggest the manuscript can be accepted for publication in the journal of Nanomaterial.